# Implicit degree bias in the link prediction task

## Abstract

Link prediction—the task of distinguishing actual hidden edges from random unconnected node pairs—is a quintessential task in graph machine learning. Despite being widely accepted as a universal benchmark and a downstream task for representation learning, its validity is seldom questioned. Here, we show that the common edge sampling procedure in link prediction introduces an implicit bias toward high-degree nodes and produces a skewed evaluation that favors methods overly reliant on node degree, to the extent that a "null" method based solely on node degree can nearly match optimal performance. To address this, we propose a degree-corrected link prediction task that offers a more accurate assessment that aligns better with performance in recommendation tasks. Finally, we demonstrate that this degree-corrected benchmark can more effectively train graph machine-learning models by reducing overfitting to node degrees and facilitating the learning of relevant structures in graphs.

## 1 Introduction

Standardized benchmarks like ImageNet (Deng et al., 2009; Krizhevsky et al., 2012) and SQuAD (Rajpurkar et al., 2016; 2018) play a pivotal role in driving progress in machine learning by fostering competition through setting clear, measurable goals. In graph machine learning, a core benchmark is link prediction, which involves identifying missing edges in a graph, with diverse applications including the recommendations of friends and contents (Kunegis & Lommatzsch, 2009; Wang et al., 2014; Huang et al., 2005; Menon & Elkan, 2011), knowledge discoveries (Sun et al., 2019; Bordes et al., 2013), and drug development (Abbas et al., 2021; Breit et al., 2020; You et al., 2019; Wang et al., 2015; Crichton et al., 2018; Yue et al., 2020; Ali et al., 2019). Link prediction benchmarks have been essential for quantitative evaluations, advancing graph machine learning techniques (Ghasemian et al., 2020; Liben-Nowell & Kleinberg, 2003; Mara et al., 2020; Breit et al., 2020; Yue et al., 2020; Ali et al., 2019; Narayanan et al., 2011).

Despite its significant role in graph machine learning, the link prediction benchmark itself is rarely scrutinized for effectiveness, reliability, and bias. Typically, it evaluates methods based on their ability to classify node pairs as connected or unconnected (Kunegis & Lommatzsch, 2009; Ghasemian et al., 2020; Mara et al., 2020). Connected pairs (edges) are randomly sampled from existing edges as the hidden positive set, while an equal number of unconnected node pairs are sampled randomly. Criticisms often highlight its disconnect from real-world scenarios. For instance, unconnected pairs vastly outnumber connected ones because of the graph sparsity (Newman, 2018; Barabási & Pósfai, 2016), leading to biased performance evaluations (Li et al., 2024a; Menand & Seshadhri, 2024; Yang et al., 2015; Huang et al., 2023; Wang et al., 2021). Additionally, the benchmark tests a predefined set of edges, while real-world tasks involve identifying potential edges across the entire graph. Despite this misalignment, high benchmark performance is often seen as a marker of successful learning in graph machine learning (Ghasemian et al., 2020; Zhang & Chen, 2018; Breit et al., 2020; Crichton et al., 2018; Yue et al., 2020; Ali et al., 2019; Grover & Leskovec, 2016; Ou et al., 2016; Goyal & Ferrara, 2018; Cai et al., 2021).

Here, we argue that the standard link prediction benchmark has a fundamental and severe bias favoring methods that exploit node degree (the number of edges a node has). This bias arises from the edge sampling process: a node with $k$ edges is $k$ times more likely to be

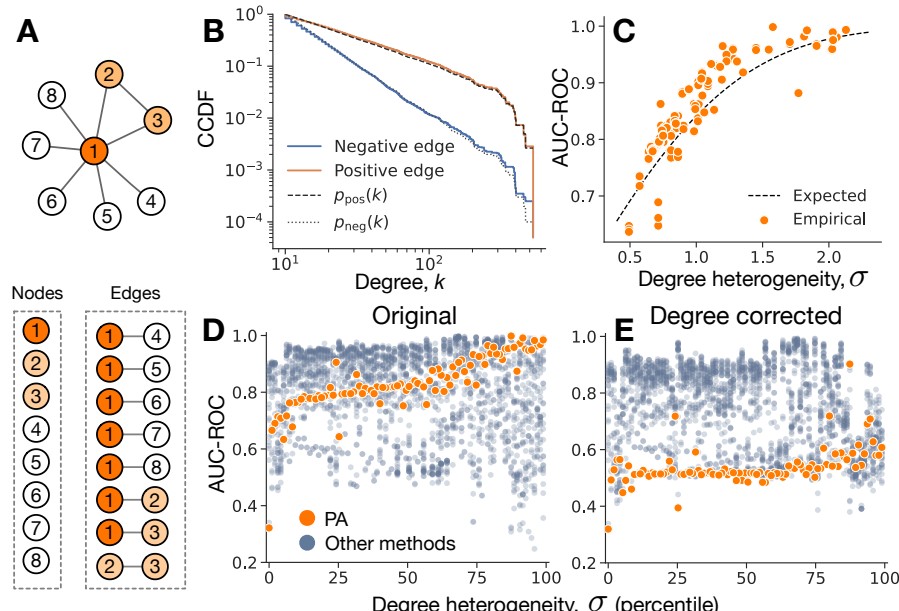

Figure 1: Illustration of the degree bias in the link prediction benchmark. A: A node with degree $k$ appears $k$ times in the edge list, making it $k$ times more likely to be sampled as a positive edge than a node with degree 1. B: The degree distribution of the nodes in the positive and negative edges sampled from a Price graph of $N = 10^5$ nodes and $M = 10^6$ edges. The y-axis, "CCDF", denotes the complementary cumulative distribution function, representing the probability that a node's degree is at least $k$. Dashed lines illustrate the relationship described by Eq. 1. C: The AUC-ROC score for the Preferential Attachment (PA) method on empirical graphs, with the dashed line indicating Eq. 4. D: AUC-ROC of 29 methods across 95 graphs. E: AUC-ROC of the same methods for the degree-corrected benchmark.

selected than a node with a single edge ($k = 1$). Meanwhile, the negative set is randomly sampled from unconnected pairs, without this degree bias. This creates a distinct feature (degree) that methods can exploit without understanding any non-trivial structural features of the graph. We show that this degree bias is so profound that a "null" method based solely on node degree can achieve near-optimal performance, questioning the benchmark's usefulness as a general objective in graph machine learning and highlighting the need for being more intentional and careful about what the evaluation tasks themselves actually evaluate.

To address this bias, we propose a degree-corrected link prediction benchmark that samples unconnected node pairs with the same degree bias. This benchmark more accurately reflects the performance of algorithms in recommendation tasks. Moreover, it trains graph neural networks more effectively by reducing overfitting to node degrees, thereby improving the learning of community structure in graphs.

## 2 Design flaw of the link prediction benchmark

### 2.1 Preliminary

We focus on unweighted, undirected graph $G = (\mathbb{V}, \mathcal{E})$, where $\mathbb{V}$ is the set of nodes and $\mathcal{E}$ is the set of edges. We assume that $G$ has no self-loops, no multiple edges, and is highly sparse ($|\mathcal{E}| \ll |\mathbb{V}|^2$), a common characteristic of real-world graphs (Newman, 2018; Barabási & Pósfai, 2016). Degree $k_i$ of a node $i \in \mathbb{V}$ is the number of edges connected to it. We use $\sim$ to denote proportional relationships. Node attributes, if present, are excluded to maintain consistency across all link prediction methods.

## 2.2 Link prediction benchmark

The standard link prediction benchmark procedure is as follows (Kunegis & Lommatzsch, 2009; Zhang & Chen, 2018; Breit et al., 2020; Crichton et al., 2018; Yue et al., 2020; Ali et al., 2019; Grover & Leskovec, 2016; Ou et al., 2016; Goyal & Ferrara, 2018; Cai et al., 2021). First, a fraction $\beta$ of edges is randomly sampled from the edge set $\mathcal{E}$ as positive edges. Second, an equal number of unconnected node pairs is randomly sampled with replacement from the node set $\mathbb{V}$ as negative edges. Negative edges are resampled if they form a loop or are already in the positive or test edges. Third, each node pair $(i, j)$ is scored by a link prediction method, where a higher score $s_{ij}$ indicates a greater likelihood of an edge. Fourth, the method's effectiveness is evaluated using the Area Under the Receiver Operating Characteristic Curve (AUC-ROC), which represents the probability that the method gives a higher score to a positive edge than a negative edge. While alternative benchmark designs use different evaluation metrics or sampling strategies for negative edges (Li et al., 2024a; Menand & Seshadhri, 2024; Yang et al., 2015; Huang et al., 2023; Wang et al., 2021; Wang & Derr, 2022; Russo et al., 2024; He et al., 2024) (Section 4), this outlined procedure is widely adopted (Kunegis & Lommatzsch, 2009; Zhang & Chen, 2018; Breit et al., 2020; Crichton et al., 2018; Yue et al., 2020; Ali et al., 2019; Grover & Leskovec, 2016; Ou et al., 2016; Goyal & Ferrara, 2018; Ghasemian et al., 2020).

## 2.3 Sampling bias due to node degree

A well-known, counterintuitive fact about graphs is that a uniform random sampling of edges introduces a degree bias in node selection (Feld, 1991; Barthélemy et al., 2004; Christakis & Fowler, 2010; Kojaku et al., 2021a;b). The bias arises because a node with $k$ edges appears $k$ times in the edge list and thus $k$ times more likely to be chosen than a node with $k' = 1$ edge (e.g., node 1 and 4 in Fig. 1A). Consequently, for a graph with degree distribution $p(k)$, the nodes in the positive edges have degree distribution $p_{\text{pos}}(k)$ proportional to $\sim k \cdot p(k)$. By normalizing $k \cdot p(k)$, the degree distribution of the positive edges is given by

$$p_{\text{pos}}(k) = \frac{1}{\sum_\ell \ell p(\ell)} k \cdot p(k) = \frac{1}{\langle k \rangle} k \cdot p(k),  \tag{1}$$

where $\langle k \rangle$ is the average degree. By contrast, nodes in the negative edges are uniformly sampled from the node set $\mathbb{V}$, resulting in a degree distribution $p_{\text{neg}}(k)$ identical to $p(k)$ (i.e., $p_{\text{neg}}(k) = p(k)$).

We demonstrate the degree bias using the Price graph with $N = 10^5$ nodes and $M = 10^6$ edges, which follows a power-law degree distribution $p(k) \propto k^{-3}$ (Fig. 1B). We uniformly sample $\beta = 0.25$ of the edges from $\mathcal{E}$ as positive edges, together with an equal number of unconnected node pairs sampled uniformly from $\mathbb{V}$. The degree distributions for nodes in the positive and negative edges align with $p_{\text{pos}}(k)$ and $p_{\text{neg}}(k)$, respectively, confirming the sampling bias due to node degree. This degree bias is not specific to the Price graph but occurs in any graph with a non-uniform degree distribution.

## 2.4 Impact of degree bias on the link prediction benchmark

We demonstrate the impact of degree bias on link prediction benchmarks (Fig.1D) by evaluating 29 link prediction methods across 95 graphs from various domains, including social, technological, informational, biological, and transportation graphs. These methods include 7 network topology-based methods (e.g., Common Neighbors (CN) (Liben-Nowell & Kleinberg, 2003)), 13 graph embedding methods (e.g., Laplacian EigenMap (EigenMap) (Belkin & Niyogi, 2003)), 2 network models (e.g., Stochastic Block Model (e.g., SBM) (Fortunato, 2010)), and 4 graph neural networks (GNNs) (e.g., Graph Convolutional Network (GCN) (Kipf & Welling, 2017)). Detailed method and graph descriptions are available in SI Section 1. We set the test edge fraction to $\beta = 0.25$ and repeat the experiment 5 times. We quantify the heterogeneity $\sigma$ of node degree by fitting a log-normal distribution to $p(k)$ and calculating its variance parameter $\sigma$. We will show that $\sigma$ is a reliable indicator of the impact of degree bias in Section 2.5.

We focus on the Preferential Attachment (PA) link prediction method, which calculates the prediction score $s_{ij} = k_i k_j$ using only node degrees. PA is a crude method that neglects key predictive features like common neighbors and shortest distance (Li et al., 2024a; Menand & Seshadhri, 2024; Lichtnwalter & Chawla, 2012; Zhang & Chen, 2018; Mao et al., 2023). However, it still outperforms most advanced methods with an average AUC-ROC of 0.84 (ranked 13th out of 29 methods; see Fig. 2A). PA performs better as the heterogeneity of node degrees increases. This outperformance is due to degree bias, where the positive edges are more likely to be formed by nodes with high degree and thereby are easily distinguishable from the negative edges. Thus, the current benchmark design favors methods that make predictions based largely on node degrees. This spurious performance of PA is evident when using Hits@K (SI Section 4.6), indicating that the issue stems from the benchmark design rather than evaluation metrics. This spurious performance of PA also persists for larger-scale graphs (SI Section 4.4).

## 2.5 Theoretical analysis

Many empirical graphs exhibit heterogeneous degree distributions, with a few nodes having exceptionally large degrees and most having small ones. These distributions are often characterized by power-law degree distribution $p(k) \propto k^{-\alpha}$ with $\alpha \in (2, 3]$ (i.e., scale-free networks) (Albert & Barabási, 2002; Barabási & Bonabeau, 2003; Holme, 2019; Voitalov et al., 2019) or log-normal distributions (Artico et al., 2020; Broido & Clauset, 2019). While the power-law and log-normal distributions are both continuous, they are often used to approximate discrete degree distribution (Artico et al., 2020; Broido & Clauset, 2019; Clauset et al., 2009; Radicchi et al., 2008; Johnson et al., 1995; Redner, 2005). We show that the AUC-ROC for PA reaches near-maximum under log-normal distributions with heterogeneous node degrees. See SI Section 4.3 for the case of power-law distributions.

Let us consider a general degree distribution $p(k)$ without restricting ourselves to log-normal distributions. The AUC-ROC has a probabilistic interpretation (Hand, 2009): it is the probability that the score $s^+$ for positive edges is larger than the score $s^-$ for negative edges. Recalling that PA computes $s_{ij} = k_i k_j$, the AUC-ROC for PA is given by

$$\text{AUC-ROC} = P(s_{i^-,j^-} \leq s_{i^+,j^+}) = P(k_{i^-} k_{j^-} \leq k_{i^+} k_{j^+}), \tag{2}$$

where $i^{\pm}$ and $j^{\pm}$ represent the nodes of the positive and negative edges, respectively. We define the degree bias by Eq 2. The AUC-ROC represents the discrepancy between the distributions of the prediction scores $k_{i^-} k_{j^-}$ for negative edges and the scores $k_{i^+} k_{j^+}$ for positive edges. If the positive edges have higher scores than the negative edges, the AUC-ROC approaches 1, indicating that node degree alone can effectively predict the positive edges. Conversely, if the positive and negative edges have similar scores, the AUC-ROC nears 0.5, indicating node degree alone is insufficient for the prediction.

Now, let us assume that $p(k)$ follows the log-normal distribution, $\text{LogNorm}(k \mid \mu, \sigma^2)$, given by (Hand, 2009):

$$p(k) = \text{LogNorm}(k \mid \mu, \sigma^2) = \frac{1}{\sqrt{2\pi}\sigma k} \exp\left[-\frac{(\ln k - \mu)^2}{2\sigma^2}\right], \tag{3}$$

where $\mu$ and $\sigma$ are the parameters of the log-normal distribution. The mean of the log-normal degree distribution is $\langle k \rangle = \exp(\mu + \sigma^2/2)$ (Hand, 2009). By leveraging a unique characteristic of log-normal distributions, we obtain the AUC-ROC for PA analytically as follows. The detailed derivation is provided in SI Section 2.

$$P(\ln s^- < \ln s^+) = 1 - \int_{-\infty}^{\infty} \text{Norm}(z^- \mid 0, 1)\Phi\left(z^- - \sqrt{2}\sigma\right) \text{dz}, \tag{4}$$

where $\Phi(z^-)$ is the cumulative distribution function for the standard normal distribution, i.e., $\Phi(z^-) = \int_{\infty}^{z^-} \text{Norm}(y \mid 0, 1)\text{d}y$. We have assumed no degree assortativity in the graph, where $P(k_i^+, k_j^+) = P(k_i)P(k_j)$. Although empirical graphs often exhibit degree assortativity, our results indicate that it does not significantly impact the AUC-ROC (SI Section 4.2).

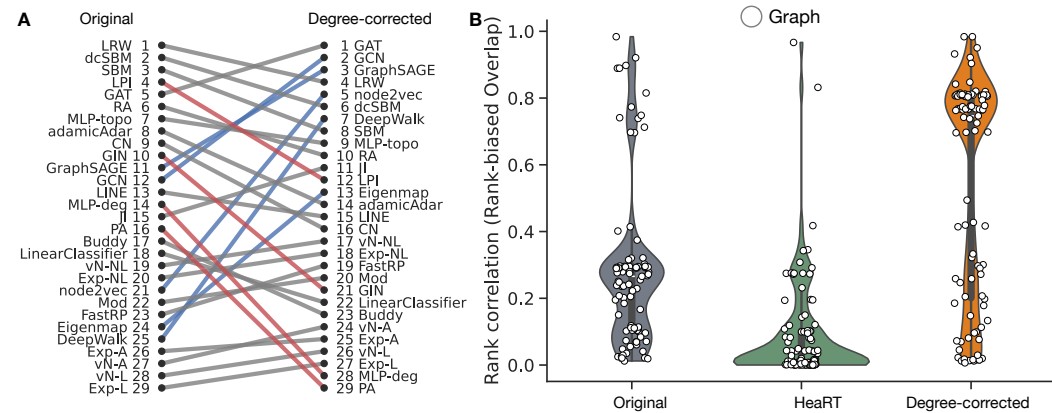

Figure 2: Comparative analysis of link prediction and recommendation benchmarks. A: Ranking changes for link prediction methods between original and degree-corrected benchmarks. Red and blue lines indicate methods with ranking shifts greater than 8 places. B: Ranking of methods by the degree-corrected benchmark is more aligned with that of the recommendation task than that of the original benchmark and the HeaRT benchmark. RBO (rank-biased overlap) measures the similarity between link prediction and recommendation task rankings.

Equation 4 suggests the key behavior of AUC-ROC for PA. The AUC-ROC for PA is an increasing function of the parameter $\sigma$ of the log-normal distribution (Fig. 1C). The parameter $\sigma$ of the log-normal distribution controls the spread of the distribution, with larger $\sigma$ resulting in a more fat-tailed distribution. While our assumptions about the log-normal degree distribution and degree assortativity may not always align with real-world data, Eq. 4 still effectively captures the AUC-ROC behavior for PA (Fig. 1C). Further analysis of power-law distributions is described in SI Section 4.3.

This theoretical result highlights the significant issue with the current link prediction benchmark: a high benchmark performance can be achieved by only learning node degrees, posing the question of whether the link prediction benchmark is an effective objective of graph machine learning.

## 3 The degree-corrected link prediction benchmark

The link prediction benchmark yields biased evaluations due to mismatched degree distributions between positive and negative edges, i.e., $p_{\text{neg}}(k) \neq p_{\text{pos}}(k)$. To mitigate the mismatch, we introduce the degree-corrected link prediction benchmark that samples negative edges with the same degree bias as positive edges (See Algorithm 1 in SI). Specifically, we create a list of nodes where each node with degree $k$ appears $k$ times. Then, we sample negative edges by uniformly sampling two nodes from this list with replacement until the sampled node pairs are not connected and not in the test edge set. Crucially, nodes with degree $k$ are $k$ times more likely to be sampled than nodes with degree 1, mirroring the degree bias of the positive edges. Consequently, the positive and negative edges in the degree-corrected benchmark are indistinguishable based on node degrees. A Python package for the degree-corrected benchmark will be made available on GitHub.

### 3.1 Comparison of the benchmark evaluations

We reevaluated the methods using the degree-corrected link prediction benchmark, maintaining the same parameters as in the original benchmark. The results show a significant drop in the performance of PA, with most methods having AUC-ROC scores close to 0.5 for most networks (Fig. 1E). We find qualitatively the same results when using the Hits@K score (SI Section 4.6).

We note that our aim is not to completely eliminate degree as a predictive feature, but rather to remove the bias introduced by negative edge sampling. The degree can remain a meaningful predictor after correction if it genuinely correlates with the likelihood of edges between nodes. The 'biokg_drug' graph effectively illustrates the precision of our degree-correction method in removing artificial bias while preserving the link predictive power of degree.

The degree-corrected benchmark has some agreement with the original benchmark in terms of the ranking of the methods (Fig. 2A). For example, they rank GAT and LRW as top performers, while NB, SGTAdjNeu, and SGTAdjExp are consistently ranked lower. On the other hand, the degree-corrected benchmark ranks PA as the lowest performer, with its average AUC-ROC dropping from 0.83 to 0.54, placing it last out of 29 methods. Other methods such as LPI, GIN also experience a substantial drop in their rankings from 4nd to 12th and 10th to 21th, respectively (Fig. 2A). On the other hand, GCN, GraphSAGE, node2vec, DeepWalk, and EigenMap increase their rankings substantially (Fig. 2A). (See SI Section 4.1 for the ranking of methods by HeaRT benchmark Li et al. (2024a).)

### 3.2   The degree-corrected benchmark aligns better with recommendation tasks

Link prediction serves as a computationally efficient proxy for evaluating and training recommendation systems (Li et al., 2024a; Menand & Seshadhri, 2024; Yang et al., 2015; Huang et al., 2023; Wang et al., 2021). In the recommendation task, directly optimizing recommendation metrics such as Hits@K requires ranking all possible node pairs for each node, which is computationally infeasible for large networks. In contrast, link prediction evaluates on a fixed set of candidate pairs, making it $\mathcal{O}(M)$ (where $M$ is the number of edges) and thus practical for both evaluation and training. This computational advantage has made link prediction benchmarks a de facto standard for developing and training recommendation models. However, this practice is only valid if link prediction performance correlates with recommendation performance. It is thus crucial that a link prediction benchmark accurately mirrors the performance in recommendation tasks.

In the recommendation task, a method must rank all potential connections for each node without a predefined candidate set. This differs fundamentally from link prediction where we evaluate on a fixed set of node pairs. Specifically, for each node $i$, a method recommends its top $C = 50$ nodes $j$ based on the highest scores $s_{ij}$. We note that our results are consistent regardless of the chosen $C$ value (refer to SI Section 4.5). The effectiveness of the recommendations is then measured using the vertex-centric max precision recall at $C$ (VCMPR@C) defined as the higher of the precision and recall scores at $C$ (Menand & Seshadhri, 2024). The VCMPR@C is designed to evaluate recommendation methods and addresses the excessive penalty on the precision scores at $C$ for small-degree nodes, which often have low precision due to their limited number of edges. We perform this task five times and average the VCMPR@C scores across different runs.

For each graph, we evaluate the alignment between the rankings based on the recommendation task and those based on the link prediction benchmarks using Rank Biased Overlap (RBO) (Webber et al., 2010). RBO is a ranking similarity metric with larger weights on the top performers in the two rankings. A larger RBO score indicates that the top performers in the two rankings are more similar. The weights on the top performer are controlled by the parameter $p \in (0, 1)$. While we set $p = 0.5$ in our experiment, we confirmed that our results are robust to the choice of $p$ (SI Section 4.5).

Our results from 95 graphs show that the degree-corrected benchmark achieves higher RBO scores than the original benchmark. For reference, we also tested the HeaRT benchmark (Li et al., 2024a), which is a recent link prediction benchmark that mitigates the distance-based bias. The results show that HeaRT achieves substantially lower RBO scores than both the original and degree-corrected benchmarks. We find consistent results for different values of $C$ and parameter $p$ of the RBO (SI Section 4.5). These results indicate that the degree-corrected benchmark more accurately mirrors the performance in recommendation tasks, providing a more reliable measure of the effectiveness of methods in practical applications.

### 3.3 Degree-corrected benchmark facilitates the learning of community structure

The link prediction benchmark is a common unsupervised learning objective for GNNs (Hamilton et al., 2017; Kawamoto et al., 2018; Kojaku et al., 2021b; Tang et al., 2015). The degree bias implies that GNNs trained using the original link prediction benchmark tend to overfit to node degrees because they can easily differentiate between positive and negative edges based on node degrees. We show that the degree-corrected benchmark effectively prevents overfitting to node degrees and improves the learning of salient graph structures.

We evaluate GNNs on the common unsupervised task of community detection in graphs (Fortunato & Newman, 2022; Fortunato, 2010; Fortunato & Hric, 2016). The community detection task identifies densely connected groups (i.e., communities) in a graph. Community detection and link prediction tasks are intimately related Clauset et al. (2008); Peixoto (2018); Ghasemian et al. (2020). Two nodes will likely have edges if they belong to the same community. By training a graph machine learning model (e.g., GNNs) to learn a node embedding to predict links, nodes in the same community are mapped to be close to each other in the embedding space Kojaku et al. (2023). Communities often correspond to functional units (e.g., social circles with similar opinions and protein complexes) in the graph, and detecting communities is a crucial task in many graph applications (Fortunato & Newman, 2022; Fortunato, 2010; Fortunato & Hric, 2016; Peixoto, 2013; 2018). Specifically, we test the GNNs by using the Lancichinetti-Fortunato-Radicchi (LFR) community detection benchmark (Lancichinetti et al., 2008), a standard benchmark for community detection (Fortunato & Hric, 2016; Fortunato, 2010; Tandon et al., 2021; Kojaku et al., 2023). The LFR benchmark generates synthetic graphs with predefined communities as follows. Each node $i$ is assigned a degree $k_i$ from a power-law distribution $p(k) \sim k^{-\tau_1}$, with maximum degree $k_{\max}$. A smaller $\tau_1$ indicates a higher likelihood of large degree nodes, which results in a more heterogeneous degree distribution. Nodes are randomly grouped into $L$ communities, with community sizes (i.e., the number of nodes in a community) following another power-law distribution $p(n) \sim n^{-\tau_2}$ bounded between $n_{\min}$ and $n_{\max}$. Edges are then formed such that each node $i$ connects to a fraction $1 - \mu$ of its $k_i$ edges within its community and the remaining fraction $\mu$ to nodes in other communities. We generate 10 graphs for $\mu \in \{0.05, 0.1, 0.15, \ldots, 0.95, 1\}$ using the following parameter values: the number of nodes $N = 3,000$, the degree exponent $\tau_1 \in \{2.5, 3\}$, the average degree $\langle k \rangle = 25$, the maximum degree $k_{\max} = 1000$, the community-size exponent $\tau_2 = 3$, the minimum and maximum community size $n_{\min} = 100$ and $n_{\max} = 1000$. We obtained qualitatively similar results for different values of the parameters of the LFR benchmark (SI Section 4.8).

We train GNNs using either the original or the degree-corrected link prediction benchmarks to minimize binary entropy loss in classifying the positive and negative edges. Using the trained GNNs, we generate node embeddings and apply the $K$-means clustering algorithm to detect communities, where $K$ is set to the number of true communities. Although the number $K$ of communities is often unknown, we use the ground-truth number to eliminate noise from estimating $K$ and to concentrate on evaluating the quality of the learned node embeddings, a standard practice in benchmarking node embeddings for the community detection task (Tandon et al., 2021; Kojaku et al., 2023; Kovács et al., 2024). We measure the performance of GNNs by comparing the detected communities against the true communities using the adjusted element-centric similarity (Kojaku et al., 2023; Kovács et al., 2024; Gates et al., 2019), where higher scores indicate a higher similarity between the node partitions for the true and detected communities. We observe qualitatively similar results for partition similarities based on the normalized mutual information (SI Section 4.7).

All GNNs, except for GIN, perform well when $\mu \leq 0.5$, where communities are distinct and easily identifiable, but their performance declines as $\mu$ increases (Fig. 3A). Across a broad range of $\mu$, degree-corrected GNNs, particularly GIN, GCN, and GraphSAGE, outperform original GNNs in identifying communities, as shown by the area under the performance curve (Fig. 3B). The advantage of degree-corrected benchmarks becomes more evident with a more heterogeneous degree distribution (Fig. 3C and D). This indicates that degree correction effectively reduces overfitting to node degrees, enhancing the learning of community structures in graphs.

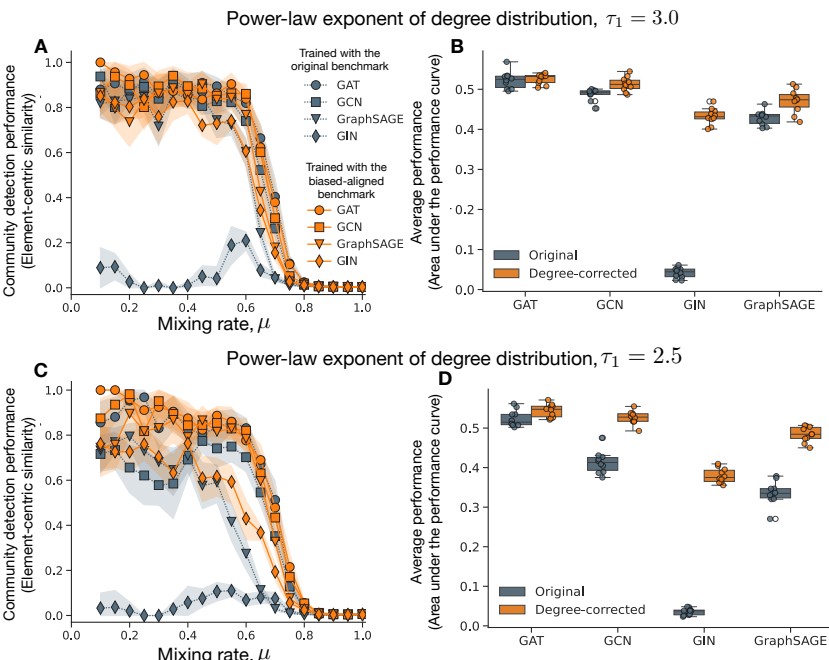

Figure 3: The degree-corrected benchmark improves GNNs in learning community structure in the LFR graphs (3000 nodes, average degree 25). A: The performance for the LFR graphs with a power-law degree distribution with $\tau_1 = 3.0$ as a function of mixing $\mu$. B: The average performance (by the area under the performance curve). C, D: The same plots for LFR graphs with $\tau_1 = 2.5$. The error bars represent the 95% confidence interval estimated by a bootstrap of 1,000 repetitions.

## 4 Discussion

We showed that common link prediction benchmarks are biased due to node degree in edge sampling, favoring methods that overfit to node degree. This bias distorts model evaluations and leads to suboptimal node embeddings (Fig. 2B, Fig. 3B). The degree bias we focused on is artifactual, i.e., it is not present in the actual network data but arises in the set of sampled positive and negative edges due to the biased sampling algorithm. We have shown that this artifactual bias significantly distorts the evaluation of link prediction models (Fig. 2B) and can be leveraged by these models to optimize their objective functions, leading to suboptimal learning of node embeddings (Fig. 3B).

To better the contribution of degree bias, we decomposed AUC-ROC scores into contributions from different node degree groups (SI Section 3). Our analysis revealed that in networks with high degree heterogeneity ($\sigma > 1.3$), a single combination—high-degree positive edges and low-degree negative edges—dominates the evaluation, accounting for over 80% of the overall AUC-ROC score. This finding explains why even simple degree-based methods perform well: the benchmark's evaluation is largely determined by cases that can be easily classified using degree alone. Indeed, our logistic regression analysis shows that in heterogeneous networks, the degree product becomes overwhelmingly important, with its coefficient more than twice as large as other structural features (SI Section 3). These results highlight how the sampling procedure inadvertently creates a shortcut that allows methods to achieve high performance without learning meaningful graph structures.

To address the degree bias, we proposed a degree-corrected benchmark that aligns the degree distributions of sampled edges. Our benchmark not only provided accurate evaluations but also improved GNN training by reducing overfitting to degree and enhancing community detection.

While our focus is on degree bias, we acknowledge other biases identified by previous studies (Li et al., 2024a; Lichtnwalter & Chawla, 2012; Zhang & Chen, 2018; Mao et al., 2023). We highlight some of these biases to underscore the uniqueness of degree bias.

(1) Distance bias arises because nodes connected by negative edges are generally farther apart than those linked by positive edges (Li et al., 2024a), making them easily distinguishable by distance. We observed that correcting for degree bias consistently reduces distance bias; the degree-corrected benchmark showed more negative edges connected by paths of length 2 compared to the standard benchmark across all networks. While our degree correction naturally mitigates distance bias, distance debiasing does not address degree bias. This is evidenced by the strong performance of PA, a purely degree-based predictor, even after distance-bias correction (SI Section 4.1). This asymmetry likely arises because node distances are inherently influenced by degree heterogeneity. In networks with high degree heterogeneity, high-degree nodes act as hubs, creating short paths between many node pairs. Correcting for degree bias naturally reduces the effect of these hub-mediated short paths. However, correcting for distance alone does not address the underlying degree heterogeneity, which continues to influence network structure and link prediction performance.

(2) Study (Huang et al., 2023) highlighted an issue arising from the substantial down-sampling of negative edges to match the number of positive edges. They proposed an "unbiased testing" approach for link prediction by evaluating methods on all possible negative pairs, rather than just a sampled subset. However, we note that the concept of "bias" in their work differs from the sampling bias we address. The degree bias we focus on primarily stems from positive edge sampling, not negative edges. Sampling all negative edges does not resolve this degree bias at all because the node frequency in all negative edges still matches that of uniformly sampled negative edges.

(3) Another potential bias could stem from overfitting to large-degree nodes, which are more frequently sampled in our degree-corrected sampling method. However, these high-degree nodes are sampled as negative examples, penalizing the model if it overfits them. This encourages the model to learn salient features of other nodes, resulting in embeddings that better capture nuanced network structures, such as community structure (Fig. 3). This effect—eliminating dominant patterns to reveal more nuanced ones—echoes the analysis of stock price time series, where filtering out the dominant market trends (e.g., recession, and inflation) can expose finer correlations between individual stocks (MacMahon & Garlaschelli,

2015). A comparable effect can also be leveraged for community detection (Newman, 2006) and node embedding (Kojaku et al., 2021b).

To better understand the mechanism of degree bias, we decomposed AUC-ROC scores into contributions from different node degree groups (SI Section 3). Our analysis revealed that in networks with high degree heterogeneity ($\sigma > 1.3$), a single combination—high-degree positive edges and low-degree negative edges—dominates the evaluation, accounting for over 70% of the overall AUC-ROC score. This finding explains why even simple degree-based methods perform well: the benchmark's evaluation is largely determined by cases that can be easily classified using degree alone. Indeed, our logistic regression analysis shows that in heterogeneous networks, the degree product becomes overwhelmingly important, with its coefficient more than twice as large as other structural features. These results highlight how the sampling procedure inadvertently creates a shortcut that allows methods to achieve high performance without learning meaningful graph structures.

In summary, our findings add a new direction to the ongoing examination of the link prediction task by demonstrating that node degree—a local and notably simpler attribute than distance—is often sufficient for differentiating the positive and negative edges. Crucially, the degree bias arises in any non-regular graph, regardless of the structure of the graph, because the bias stems not from the graph structure but from the edge sampling algorithm used in the link prediction benchmark. More broadly, edge sampling—the source of the degree bias—is a general technique for evaluating and training graph machine learning. For instance, mini-batch training (Hamilton et al., 2017; Hu et al., 2020), which samples subsets of edges for efficient GNN training, may also exhibit bias due to node degrees, leading to skewed training sets. Given the widespread use of edge sampling across various graph machine-learning tasks, our findings have broad implications beyond link prediction benchmarks, extending to a range of benchmarks and training frameworks.

Our study has several limitations. First, we did not explore the reasons behind the varying performance of different link prediction methods. Degree heterogeneity can negatively impact GNNs (Wang et al., 2023; Liu et al., 2021; Subramonian et al., 2023; Li et al., 2024b; Liu et al., 2023; Subramonian et al., 2024; Kang et al., 2022; Arun et al., 2023). Small-degree nodes tend to have poor representation quality due to limited neighborhood information (Wang et al., 2023; Liu et al., 2021), and large-degree nodes benefit from reinforced structural inequality (Subramonian et al., 2023; Li et al., 2024b; Liu et al., 2023). This—how well a method represents low-degree nodes—could be one reason for the differences in performance. It is important to note that there is no single link prediction method that is universally effective for all graphs because the performance of link prediction methods depends on the assumptions on the graph structure to make the predictions (Ghasemian et al., 2020). Second, we focus on community structure to test the effectiveness of the proposed benchmark as a training framework. However, other non-trivial graph structures, such as centrality, could be tested through network dismantling benchmarks (Osat et al., 2023). Third, our degree correction method addresses one form of bias, it may potentially introduce new biases, although we could not identify any clear example of such biases. As a precaution, we investigated whether our debiasing method can exacerbate certain other biases identified in literature such as the distance bias Li et al. (2024a) and found no evidence of this. In fact, we find that our method mitigates the distance bias not vice-versa. Fourth, our analysis focuses on the transductive setting, where link prediction occurs between nodes in the training graph. We note that degree bias likely persists inductively since new nodes with more connections are still more likely to be selected in edge sampling, mirroring the bias in the transductive setting. Fifth, we focused on undirected and undirected graphs, which are also common. It is trivial to extend our results to directed graphs, where the degree bias is now dependent on both the out-degree and in-degree of nodes. For weighted graphs, if the weight represent multiplicities of edges, our degree-corrected sampling method can be applied straightforwardly.

Despite these limitations, our results suggest that sampling graph data is a highly non-trivial task than commonly considered. Because sampling edges from a graph is integral to evaluating and training graph machine learning methods, our results underline the importance of careful sampling to ensure the effectiveness of evaluations and training of graph machine learning methods.

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

## A  Appendix

You may include other additional sections here.

# Supplementary Information for "Implicit degree bias in the link prediction task"

Anonymous authors

## Contents

# 1  Data and link prediction methods

## 1.1  Network data

The corpus of networks used in this work comprises networks with the number of nodes in the range $[10^2, 10^6]$ and edges in the range $[10^2, 10^8]$. We also considered networks from the OGB benchmark Hu et al. (2020). This includes social, technological, information, biological, and transportation (spatial) networks. For simplicity in our analysis, we consider these networks to be unweighted, undirected, and without self-loops, though the message of our work holds without these constraints. The largest networks in our corpus (number of nodes $> 10^5$) are sourced from Netzschleuder (Peixoto, 2020), and the remaining networks are obtained from the authors of Ref. (Erkol et al., 2019). See Table 2 for details.

## 1.2  Link prediction algorithms

We use 26 link prediction algorithms categorized into four groups: topology-based, graph embedding, network model, and graph neural networks (see Table 1).

### 1.2.1  Topology-based predictors

Topology-based predictors calculate the prediction score $s_{ij}$ using the structural features of two nodes. The topology-based predictors employed in our study include Preferential Attachment (PA) (Barabási & Pósfai, 2016), Common Neighbors (CN) (Liben-Nowell & Kleinberg, 2003), Adamic-Adar (AA) (Adamic & Adar, 2003), Jaccard Index (JI) (Liben-Nowell & Kleinberg, 2003), Resource Allocation (RA) (Zhou et al., 2007; 2009), Local Random Walk (LRW) (Liu & Lü, 2010), Local Path Index (LPI) (Lü et al., 2009). For LRW and LPI, we set the hyperparameter $\epsilon = 0.001$ as per previous studies (Lü et al., 2009; Liu & Lü, 2010). The other methods do not require hyperparameters.

We implemented two multilayer perceptrons (MLPs) that takes features of two nodes and predict whether they are connected by an edge or not. The first MLP (MLP-deg) takes only the degree features as input, i.e., degree product $k_i k_j$, degree sum $k_i + k_j$, minimum degree $\min(k_i, k_j)$, and maximum degree $\max(k_i, k_j)$. The second MLP (MLP-topo) takes AA, JI, RA, and LRW as input. The MLP consists of two hidden layers coupled with a LeakyReLU activation function, and we used held-out validation to tune the number of dimensions in each hidden layer (32, 64) and dropout rate (0.2, 0.5) with the validation set consisting of 10% of the edges. We used the Adam optimizer at a learning rate 0.001.

As a simpler baseline, we also implemented a logistic regression model (Linear) that takes the concatenation of the node features as input and predict whether they are connected by an edge or not. The input features are RA, JI, and LRW, and AA, and PA. In order to reduce the collinearity between the input features, we performed feature orthogonalization by regressing RA, JI, and LRW, AA on PA and taking the residuals as new features. This means that after orthognalizations, we input the residuals of RA, JI, and LRW, AA on PA, as well as the raw PA features to the logistic regression model. To further reduce the collinearity, we employed the ridge regression implemented in scikit-learn (Pedregosa et al., 2011), with the regularization parameter set to the default value.

### 1.2.2  Graph embeddings

Graph embedding maps a graph into a vector space, with each node $i$ represented by a point in this space. The prediction score $s_{ij}$ is given by the dot product $\vec{u}_i^\top \vec{u}_j$ between any two node vectors. We tested a variety of graph embedding methods including Laplacian EigenMap (EigenMap) (Belkin & Niyogi, 2003), Spectral Modularity (Mod) (Nadakuditi & Newman, 2012), Non-backtracking Embedding (NB) (Krzakala et al., 2013), FastRP (FastRP) (Chen et al., 2019), Exponential Kernel on Adjacency Matrix (Exp-A) (Kondor & Lafferty, 2002; Kunegis & Lommatzsch, 2009), Exponential Kernel on Laplacian (Exp-L) (Kondor & Lafferty, 2002; Kunegis & Lommatzsch, 2009), Exponential Kernel on Normalized Laplacian

(Exp-NL) (Kondor & Lafferty, 2002; Kunegis & Lommatzsch, 2009), Von Neumann Kernel on Adjacency Matrix (vN-A) (Ito et al., 2005; Kunegis & Lommatzsch, 2009), Von Neumann Kernel on Laplacian (vN-L) (Ito et al., 2005; Kunegis & Lommatzsch, 2009), and Von Neumann Kernel on Normalized Laplacian (vN-NL) (Ito et al., 2005; Kunegis & Lommatzsch, 2009), node2vec (node2vec) (Grover & Leskovec, 2016), DeepWalk (DeepWalk) (Perozzi et al., 2014), and LINE (LINE) (Tang et al., 2015). We tested a variety of graph embedding methods including Laplacian EigenMap (EigenMap) (Belkin & Niyogi, 2003), Spectral Modularity (Mod) (Nadakuditi & Newman, 2012), Non-backtracking Embedding (NB) (Krzakala et al., 2013), FastRP (FastRP) (Chen et al., 2019), Exponential Kernel on Adjacency Matrix (Exp-A) (Kondor & Lafferty, 2002; Kunegis & Lommatzsch, 2009), Exponential Kernel on Laplacian (Exp-L) (Kondor & Lafferty, 2002; Kunegis & Lommatzsch, 2009), Exponential Kernel on Normalized Laplacian (Exp-NL) (Kondor & Lafferty, 2002; Kunegis & Lommatzsch, 2009), Von Neumann Kernel on Adjacency Matrix (vN-A) (Ito et al., 2005; Kunegis & Lommatzsch, 2009), Von Neumann Kernel on Laplacian (vN-L) (Ito et al., 2005; Kunegis & Lommatzsch, 2009), and Von Neumann Kernel on Normalized Laplacian (vN-NL) (Ito et al., 2005; Kunegis & Lommatzsch, 2009), node2vec (node2vec) (Grover & Leskovec, 2016), DeepWalk (DeepWalk) (Perozzi et al., 2014), and LINE (LINE) (Tang et al., 2015). For all methods, we set the number of embedding dimensions to 128. For LINE, node2vec, and DeepWalk, we set the number of walkers to 40 and the number of the walk length to 80 following Ref. (Kojaku et al., 2023). We used the default hyperparameters used in the original papers unless otherwise specified.

### 1.2.3 Graph Neural Networks

Graph neural networks (GNNs) learn the vector representation, $\vec{u}_i$, for each node $i$ of the network by using neural networks. The prediction score $s_{ij}$ is given by the dot product $\vec{u}_i^\top \vec{u}_j$ between any two node vectors. We also explore several graph neural network (GNN) architectures for link prediction, leveraging the PyTorch Geometric library (Fey & Lenssen, 2019). The GNN methods we employ include: Graph Convolutional Network (GCN) (Kipf & Welling, 2017), Graph SAGE (GraphSAGE) (Hamilton et al., 2017), Graph Attention Network (GAT) (Veličković et al., 2018), and Graph Isomorphism Network (GIN) (Xu et al., 2018). We used held-out validation to tune the number of hidden layers (1 or 2) and the number of dimensions in each hidden layer (64, 128, or 256) with the validation set consisting of 10% of the edges. We use ReLu activation and dropout rate of 0.2. The node features are the 64 principal eigenvectors of the adjacency matrix, and we extend the feature vector by adding a 64-dimensional vector with each element being generated from an independent Gaussian distribution with mean 0 and standard deviation 1 by following Ref. (Sato et al., 2021; Abboud et al., 2020). We train GNNs on the link prediction task for 250 epochs with a dropout rate of 0.2, using the Adam optimizer at a learning rate 0.01. We use the 'LinkNeighborLoader' from PyTorch Geometric to generate training mini-batches. This loader samples both positive and negative edges, along with 30 immediate neighbors and 10 secondary neighbors sampled by random walks for each node involved in these edges (Hamilton et al., 2017). The batch size is set to 5000.

We also tested BUDDY GNN that achieves a competitive performance on the link prediction task (Chamberlain et al., 2022). We employed hyperparameter tuning for the number of hidden channels (256 or 1024), and the feature dropout rate (0.05 or 0.2) with the validation set consisting of 10% of the edges. We set the number of hops to 2 because it consistently achieved a better performance. For other hyperparameters, we used the default values used in the original implementation.

### 1.2.4 Nework models

We use two stochastic block models (SBM) (Fortunato, 2010; Fortunato & Hric, 2016; Fortunato & Newman, 2022; Peixoto, 2013) and the degree-corrected SBM (Karrer & Newman, 2011; Peixoto, 2013; Gra). These models estimate the probability $P(i, j)$ that an edge exists between two nodes, which serves as the prediction score $s_{ij}$. We fit the SBMs using the graph tool package (Gra). We select the number of blocks that minimize the description length and use default settings for other parameters.

Table 1: Link prediction algorithms. "pyg" refers to PyTorch Geometric.

| | Algorithm | Reference | Code | Notation |
|---|---|---|---|---|
| Topology based | Preferential attachment | (Barabási & Pósfai, 2016) | ourselves | PA |
| | Common neighbors | (Liben-Nowell & Kleinberg, 2003) | ourselves | CN |
| | AdamicAdar | (Adamic & Adar, 2003) | ourselves | AA |
| | Jaccard index | (Liben-Nowell & Kleinberg, 2003) | ourselves | JI |
| | Resource allocation | (Zhou et al., 2007; 2009) | ourselves | RA |
| | Local Random Walk | (Liu & Lü, 2010) | ourselves | LRW |
| | Local Path Index | (Lü et al., 2009) | ourselves | LPI |
| | MLP-deg | | ourselves | MLP-deg |
| | MLP-topo | | ourselves | MLP-topo |
| | Linear model | | ourselves | Linear |
| Graph embedding | Laplacian EigenMap | (Belkin & Niyogi, 2003) | ourselves | EigenMap |
| | Spectral modularity | (Nadakuditi & Newman, 2012) | ourselves | Mod |
| | Non-backtracking embedding | (Krzakala et al., 2013) | ourselves | NB |
| | FastRP | (Chen et al., 2019) | ourselves | FastRP |
| | Adjacency matrix w/ the exponential kernel | (Kondor & Lafferty, 2002), (Kunegis & Lommatzsch, 2009) | ourselves | Exp-A |
| | Laplacian w/ the exponential kernel | (Kondor & Lafferty, 2002), (Kunegis & Lommatzsch, 2009) | ourselves | Exp-L |
| | Normalized Laplacian w/ the exponential kernel | (Kondor & Lafferty, 2002), (Kunegis & Lommatzsch, 2009) | ourselves | Exp-NL |
| | Adjacency matrix w/ the von Neumann kernel | (Kondor & Lafferty, 2002), (Kunegis & Lommatzsch, 2009) | ourselves | vN-A |
| | Laplacian w/ the von Neumann kernel | (Ito et al., 2005), (Kunegis & Lommatzsch, 2009) | ourselves | vN-L |
| | Normalized Laplacian w/ the von Neumann kernel | (Ito et al., 2005), (Kunegis & Lommatzsch, 2009) | ourselves | vN-NL |
| | LINE | (Tang et al., 2015) | gensim (Rehurek & Sojka, 2011), (Abraham, 2020) | LINE |
| | DeepWalk | (Perozzi et al., 2014) | gensim (Rehurek & Sojka, 2011), gensim (Abraham, 2020) | DeepWalk |
| | node2vec | (Grover & Leskovec, 2016) | gensim (Rehurek & Sojka, 2011), gensim (Abraham, 2020) | node2vec |
| Graph neural networks | Graph Convolutional Network | (Kipf & Welling, 2017) | pyg (Fey & Lenssen, 2019) | GCN |
| | Graph SAGE | (Hamilton et al., 2017) | pyg (Fey & Lenssen, 2019) | GraphSAGE |
| | Graph Attention Network | (Veličković et al., 2018) | pyg (Fey & Lenssen, 2019) | GAT |
| | GIN | (Xu et al., 2018) | pyg (Fey & Lenssen, 2019) | GIN |
| | BUDDY GNN | (Chamberlain et al., 2022) | pyg (Fey & Lenssen, 2019) | BUDDY |
| Network model | Stochastic block model | (Fortunato, 2010; Fortunato & Hric, 2016), (Fortunato & Newman, 2022; Peixoto, 2013) | graph tool (Gra) | SBM |
| | Degree-corrected stochastic block model | (Karrer & Newman, 2011; Peixoto, 2013; Gra) | graph tool (Gra) | dcSBM |

### 1.3 Pseudo-code for the degree-corrected link prediction benchmark

The pseudo-code for the degree-corrected link prediction benchmark is shown in Table. 1.

---

**Algorithm 1** Degree-corrected link prediction benchmark

1: Input: Graph $G(\mathbb{V}, \mathcal{E})$, Sampling fraction $\beta \in [0, 1]$ for positive edges
2: Output: Set of negative edges $\mathcal{E}_{neg}$ and set of positive edges $\mathcal{E}_{pos}$
3: Generate $\mathcal{E}_{pos}$ by randomly sampling $\beta$ fraction of edges in $\mathcal{E}$.
4: Initialize $\mathcal{E}_{neg} \leftarrow \emptyset$
5: Create a node list $L$ where each node $i \in \mathbb{V}$ with degree $k_i$ appears $k_i$ times
6: while $|\mathcal{E}_{neg}| < |\mathcal{E}_{pos}|$ do
7:     Randomly select two nodes $i, j$ from $L$ with replacement
8:     if $(i, j) \notin \mathcal{E}$ and $(i, j) \notin \mathcal{E}_{neg}$ and $i \neq j$ then
9:         $\mathcal{E}_{neg} \leftarrow \mathcal{E}_{neg} \cup \{(i, j)\}$
10:     end if
11: end while
12: return $\mathcal{E}_{neg}, \mathcal{E}_{pos}$

---

### 1.4 Additional analysis methods

We use the following additional analysis methods in this paper. We fit a log-normal distribution to the degree distribution of the graphs by using the moment method implemented in the scipy.stats.lognorm package (Virtanen et al., 2020). We fit a power-law distribution to the degree distribution of the graphs by using the maximum likelihood method implemented in the powerlaw package (Alstott et al., 2014). We compute the RBO score by using the rbo package (rbo).

## 2 AUC-ROC for the preferential attachment method

Let us first derive the degree distribution of nodes in the positive edges. By substituting Eq. 3 into Eq. 1 in the main text, we derive the degree distribution of nodes in the positive edges as:

$$
\begin{aligned}
p_{\text{pos}}(k) &= \frac{k}{\langle k \rangle} \frac{1}{\sqrt{2\pi}\sigma k} \cdot \exp\left[-\frac{(\ln k - \mu)^2}{2\sigma^2}\right] \\
&= \frac{1}{\sqrt{2\pi}\sigma k} \exp\left[-\frac{1}{2\sigma^2}\left((\ln k)^2 - 2\mu \ln k + \mu^2\right) + \ln k - \mu - \sigma^2/2\right] \\
&= \frac{1}{\sqrt{2\pi}\sigma k} \exp\left[-\frac{1}{2\sigma^2}\left((\ln k)^2 - 2(\mu + \sigma^2) \ln k + \mu^2 + 2\mu\sigma^2 + \sigma^4\right)\right] \\
&= \frac{1}{\sqrt{2\pi}\sigma k} \exp\left[-\frac{(\ln k - \mu - \sigma^2)^2}{2\sigma^2}\right] = \text{LogNorm}(k \mid \mu + \sigma^2, \sigma^2).
\end{aligned}
\tag{1}
$$

Equation 1 indicates that the degree distribution for nodes in the positive edges also follows a log-normal distribution, parameterized by $\mu + \sigma^2$ and $\sigma$.

We derive the AUC-ROC for PA by leveraging a unique characteristic of log-normal distributions, i.e., the logarithm $\ln k$ of log-normally-distributed degree $k$ follows a normal distribution with mean $\mu$ and variance $\sigma^2$, i.e.,

$$
P(\ln k) = \text{Norm}(k \mid \mu, \sigma^2), \text{ where } \text{Norm}(k \mid \mu, \sigma^2) = \frac{1}{\sqrt{2\pi}\sigma} \exp\left[-\frac{(k - \mu)^2}{2\sigma^2}\right].
\tag{2}
$$

We assume no degree assortativity in the graph, where $P(k_i^+, k_j^+) = P(k_i)P(k_j)$. Although empirical graphs often exhibit degree assortativity, our results indicate that it does not significantly impact the AUC-ROC (SI Section 4.2). For the negative edges, the distribution for $\ln s_{i^-,j^-} = \ln k_i^- + \ln k_j^-$ follows a normal distribution with mean $2\mu$ and variance $2\sigma^2$, as the sum of independent normal variables also forms a normal distribution with additive means and variances (Bishop, 2006), i.e.,

$$
P(\ln s_{i^-,j^-}) = \text{Norm}\left(\ln s_{i^-,j^-} \mid 2\mu, 2\sigma^2\right).
\tag{3}
$$

For the positive edges, the degree distribution also follows a log-normal distribution (Eq. 1). Thus, the distribution for $\ln s_{i^+,j^+} = \ln k_i^+ + \ln k_j^+$ is given by

$$
P(\ln s_{i^+,j^+}) = \text{Norm}\left(\ln s_{i^+,j^+} \mid 2\mu + 2\sigma^2, 2\sigma^2\right).
\tag{4}
$$

Thus, we have

$$
\begin{aligned}
\text{AUC-ROC} &= P(\ln s^- < \ln s^+) \\
&= \int_{-\infty}^{\infty} \text{Norm}(x^- \mid 2\mu, 2\sigma^2) \left[1 - \int_{-\infty}^{x^-} \text{Norm}(x^+ \mid 2\mu + 2\sigma^2, 2\sigma^2) \mathrm{d}x^+\right] \mathrm{d}x^- \\
&= 1 - \int_{-\infty}^{\infty} \text{Norm}(x^- \mid 2\mu, 2\sigma^2) \int_{-\infty}^{x^-} \text{Norm}(x^+ \mid 2\mu + 2\sigma^2, 2\sigma^2) \mathrm{d}x^+ \mathrm{d}x^-
\end{aligned}
\tag{5}
$$

We reparameterize Eq. 5 by using $z^{\pm} = \frac{x^{\pm} - 2\mu}{\sqrt{2}\sigma}$. Noting that $\text{Norm}(x^- \mid 2\mu, 2\sigma^2) \cdot \sqrt{2}\sigma = \text{Norm}(z^- \mid 0, 1)$ and $\mathrm{d}x^{\pm} = (\sqrt{2}\sigma)\mathrm{d}z^{\pm}$, we have

$$
\begin{aligned}
P(\ln s^- < \ln s^+) &= 1 - \int_{-\infty}^{\infty} (2\sigma^2)\text{Norm}(x^- \mid 2\mu, 2\sigma^2) \int_{-\infty}^{x^-} \text{Norm}(x^+ \mid 2\mu + 2\sigma^2, 2\sigma^2) \cdot \mathrm{d}z^+ \mathrm{d}z^- \\
&= 1 - \int_{-\infty}^{\infty} \text{Norm}\left(z^- \mid 0, 1\right) \int_{-\infty}^{z^-} \text{Norm}\left(z^+ - \sqrt{2}\sigma \mid 0, 1\right) \mathrm{d}z^+ \mathrm{d}z^- \\
&= 1 - \int_{-\infty}^{\infty} \text{Norm}(z^- \mid 0, 1) \Phi\left(z^- - \sqrt{2}\sigma\right) \mathrm{d}z,
\end{aligned}
\tag{6}
$$

where $\Phi(z^-)$ is the cumulative distribution function for the standard normal distribution, i.e., $\Phi(z^-) = \int_{\infty}^{z^-} \text{Norm}(y \mid 0, 1)\mathrm{d}y$.

Table 2: Mean absolute coefficients from logistic regression analysis across different graphs. Features were orthogonalized to eliminate collinearity, and coefficients were normalized by feature L2 norm. Larger coefficients indicate greater importance in link prediction.

| | Mean Absolute Coefficient | | |
| --- | --- | --- | --- |
| Topological Feature | All Graphs | Graphs with $\sigma > 1$ | Graphs with $\sigma > 1.5$ |
| Random walk | 11.52 | 10.68 | 9.01 |
| Degree product | 8.79 | 17.19 | 28.00 |
| Resource allocation | 2.81 | 4.53 | 6.88 |
| Adamic-Adar | 1.35 | 2.35 | 2.89 |
| Jaccard index | 0.89 | 1.06 | 0.17 |

## 3 Decomposition analysis of AUC-ROC scores

A key concern with the standard benchmark is that link prediction methods may overfit to nodes with high degrees. To investigate this systematically, we developed a decomposition analysis of the AUC-ROC scores that reveals how different groups of nodes contribute to the overall performance metrics.

The AUC-ROC score—the probability that a positive sample has a higher score than a negative sample—can be decomposed into conditional scores by partitioning evaluation edges into groups. Specifically, if we partition edges into groups $g_1$ and $g_2$, the AUC-ROC score can be written as:

$$P(s^+ > s^-) = \sum_{\ell=1}^{2} \sum_{\ell'=1}^{2} P(s_i^+ > s_j^- | i \in g_\ell^+, j \in g_{\ell'}^-) P(i \in g_\ell^+) P(j \in g_{\ell'}^-) \qquad (7)$$

where $s_i^+$ and $s_j^-$ are the scores of the positive and negative edges given by a link prediction method, respectively, and $g_\ell^+$ and $g_\ell^-$ are the positive and negative edges in the $\ell$-th group, respectively. Probability $P(s_i^+ > s_j^- | i \in g_\ell^+, j \in g_{\ell'}^-)$ represents the conditional AUC-ROC score for a positive edge sampled from $g_\ell^+$ and a negative edge sampled from $g_{\ell'}^-$. Probability $P(i \in g_\ell^+) P(j \in g_{\ell'}^-)$ represents the probability that a positive edge is sampled from $g_\ell^+$ and a negative edge is sampled from $g_{\ell'}^-$. We note that $P(i \in g_\ell^+) P(j \in g_{\ell'}^-)$ is determined by the sampling of positive and negative edges and independent of the link prediction methods.

Based on this decomposition, we investigate the impact of edges of different node degrees on the overall AUC-ROC scores. Specifically, we partition edges into two equal-sized groups based on the degree product $z = k_i k_j$ of their endpoint nodes, with $g_1$ having $z \geq$ median and $g_2$ having $z <$ median. We tested multiple definitions of $z$ including degree sum ($k_i + k_j$), minimum degree ($\min(k_i, k_j)$), and maximum degree ($\max(k_i, k_j)$), finding consistent results across all definitions.

To measure how uniformly different groups contribute to the AUC-ROC score, we use normalized entropy:

$$H = -\sum_{\ell=1}^{2} \sum_{\ell'=1}^{2} P(i \in g_\ell^+) P(j \in g_{\ell'}^-) \log P(i \in g_\ell^+) P(j \in g_{\ell'}^-) / \log 4 \qquad (8)$$

The entropy is bounded between 0 and 1, where 0 indicates that the AUC-ROC score is determined by a single group pair and 1 indicates that the AUC-ROC score is evenly distributed across all group pairs.

We observe that the standard benchmark exhibits notable disparity compared to HeaRT and degree-corrected benchmark, indicating that the contribution to the AUC-ROC score in the standard benchmark can be heavily skewed toward certain combinations of node degrees

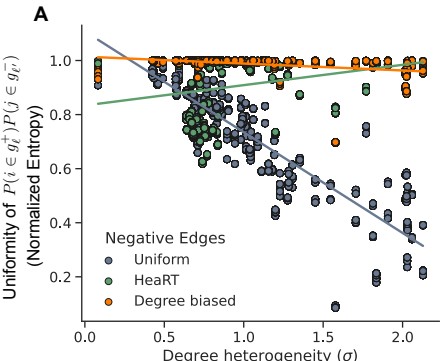 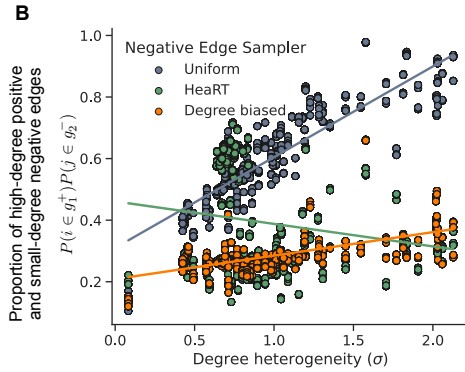

Figure 1: Decomposition of the AUC-ROC score by degree combinations. A: Uniformity of $P(i \in g_\ell^+)P(j \in g_{\ell'}^-)$ quantified by the normalized entropy, $H$. B: Probability $P(i \in g_1^+)P(j \in g_2^-)$ that the positive edges are sampled from the high-degree node group $g_1^+$ and the negative edges are sampled from the low-degree node group $g_2^-$.

(Fig. 1A). This disparity becomes even more pronounced in networks with high degree heterogeneity ($H < 0.5$ and $\sigma > 1.5$). This disparity is caused by a single group pair—positive edges from high-degree nodes and negative edges from low-degree nodes——with $P(i \in g_1^+)P(j \in g_2^-) > 0.7$ for $\sigma > 1.5$ (Fig. 1B). This means that for highly heterogeneous networks, more than 70% of the AUC-ROC score is determined by cases that can be easily classified using degree alone. The degree-corrected benchmark achieves high uniformity ($H \approx 1$) across group pairs (Fig. 1A), indicating that the contribution to the AUC-ROC score is approximately evenly distributed across different degree groups.

To further validate these findings, we performed additional experiments using logistic regression to analyze feature importance. The model was trained with resource allocation, Jaccard index, Adamic Adar, local random walk, and degree product. To ensure fair comparison, we orthogonalized the non-degree features with respect to degree to eliminate collinearity effects. Additionally, we use ridge regularization to further mitigate the effect of collinearity. We use the scikit-learn package (Pedregosa et al., 2011) to perform the logistic regression with the default ridge regularization strength. To make the regression coefficients comparable, we normalize the features by their L2 norm before training the model. The results showed that in networks with high degree heterogeneity ($\sigma > 1$), the degree product emerges as significantly important, with its coefficient 1.7 3 times as large as the second most important feature (Table. 2).

This dominance of degree-based prediction is particularly concerning because it indicates that learning-based methods can achieve high benchmark performance by primarily exploiting degree information rather than learning more complex structural patterns. This "shortcut" is precisely what our degree-corrected benchmark aims to prevent.

These results provide strong quantitative support for our argument that the standard benchmark's evaluation is dominated by easily-classified degree-based cases, potentially leading to suboptimal model training. The degree-corrected benchmark successfully addresses this issue by ensuring more uniform contributions from different degree groups, leading to a more meaningful evaluation of link prediction methods.

## 4 Robustness analysis

### 4.1 Method ranking by HeaRT benchmark

The ranking of methods by the HeaRT benchmark is shown in Fig. 2. The results show that PA achieves the highest AUC-ROC scores among all methods in the HeaRT benchmark, contrasting sharply with its lowest performance in the degree-corrected benchmark. This

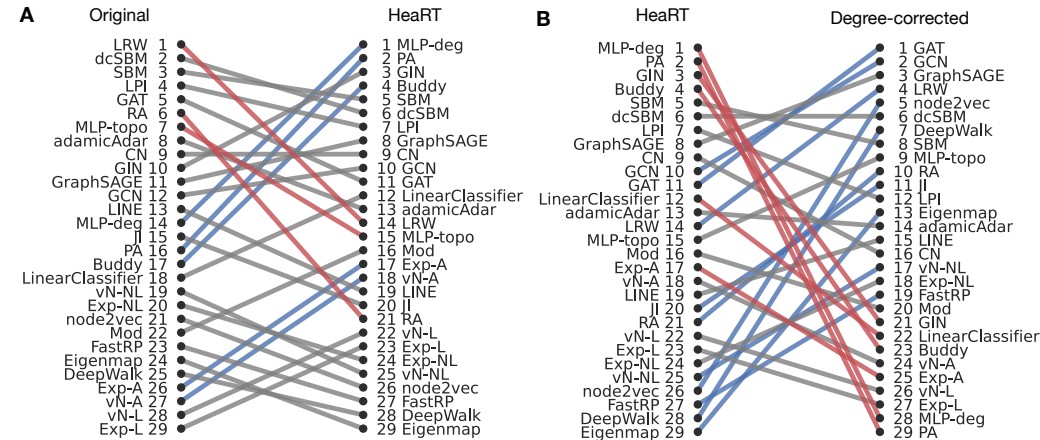

Figure 2: Comparison of the AUC-ROC scores between the original, degree-corrected, and the HeaRT benchmarks.

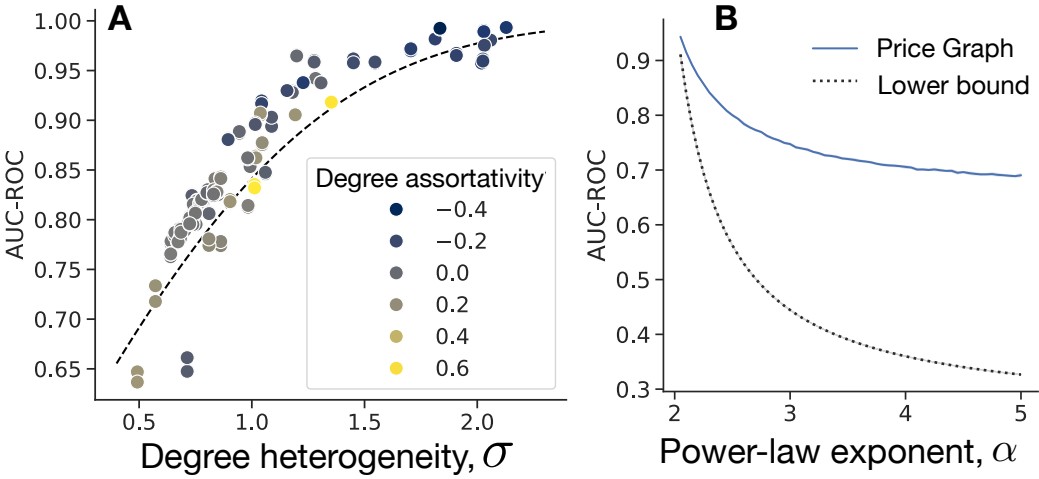

Figure 3: The AUC-ROC for PA as a function of degree heterogeneity. A: The AUC-ROC for the empirical graphs and that expected by node degree (Eq. (9) in the main text). The colors represent the degree assortativity. B: Lower bound for the AUC-ROC for the power-law degree distributions. The dashed line represents the lower bound for the AUC-ROC for the power-law distribution. The blue line represents the AUC-ROC for PA for the Price graph with $N = 10^4$ nodes and $M = 10^5$ edges.

indicates that degree bias may not be reduced by HeaRT that reduces the distance-based bias.

## 4.2 Impact of degree assortativity on the AUC-ROC for PA

We have assumed that the graph has no degree assortativity, meaning that $P(k_i, k_j) = P(k_i)P(k_j)$. Although this assumption may not always hold, it provides a good approximation for the AUC-ROC behavior for PA. Although the assortativity varies across graphs, the AUC-ROC for PA still closely follows Eq. (9) in the main text (Fig. 3).

### 4.3 AUC-ROC for PA for scale-free networks

We have assumed that the graph exhibits the heterogeneous degree distributions characterized by the log-normal distribution. An alternative model of the degree distribution is the power-law distribution (Barabási & Bonabeau, 2003). Here, we show that our results also hold for the power-law degree distribution, i.e., the AUC-ROC for PA increases as the degree heterogeneity increases.

We compute the AUC-ROC for PA for graphs with power-law degree distribution. Computing AUC-ROC $P(k_{i-}k_{j-} \leq k_{i+}k_{j+})$ is not trivial because it involves multiplicative convolution of two probability distributions, which are hard to compute for the power law degree distribution. To circumvent this problem, we consider the lower-bound by focusing on $k_{i-} \leq k_{i+}$ and $k_{j-} \leq k_{j+}$, which is the subset of all combinations of $(k_{i-}, k_{i+}, k_{j-}, k_{j+})$ leading to $k_{i-}k_{j-} \leq k_{i+}k_{j+}$, i.e.,

$$P(k_{i-} < k_{i+}) \cdot P(k_{j-} < k_{j+} \mid k_{i-}, k_{i+}) \leq P(k_{i-}k_{j-} < k_{i+}k_{j+}) \tag{9}$$

Assuming that the graph has no degree assortativity (i.e., $P(k_i, k_j) = P(k_i)P(k_j)$), we obtain the lower bound for the AUC-ROC:

$$P(k_{i-} < k_{i+}) \geq P(k_{i-} < k_{i+})^2 = \left[ \sum_{k=1}^{\infty} p_{\text{neg}}(k) \sum_{\ell=k}^{\infty} p_{\text{pos}}(\ell) \right]^2. \tag{10}$$

Now, let us compute the lower bound by assuming that the degree distribution follows a power-law (Clauset et al., 2009):

$$p(k) = \frac{1}{\zeta(\alpha, k_{\min})} k^{-\alpha}, \;\; (k \geq k_{\min}), \text{ where } \zeta(\alpha, k_{\min}) = \sum_{\ell=k_{\min}}^{\infty} \ell^{-\alpha}, \tag{11}$$

where $\zeta$ is the Hurwitz zeta function, and $k_{\min}$ is the minimum degree. By substituting Eq. (1) in the main text into Eq. equation 11, we have $p_{\text{pos}} = k^{-\alpha+1}/\zeta(\alpha - 1, k_{\min})$. By noting that $\sum_{\ell=k}^{\infty} p(\ell) = \zeta(\alpha, k)/\zeta(\alpha, k_{\min})$ (Clauset et al., 2009), we have

$$P(k_{i-} < k_{i+})^2 = \left[ \frac{1}{\zeta(\alpha, k_{\min})\zeta(\alpha - 1, k_{\min})} \sum_{k=k_{\min}}^{\infty} k^{-\alpha}\zeta(\alpha - 1, k) \right]^2. \tag{12}$$

Numerical calculation shows that the lower bound $P(k_{i-} < k_{i+})^2$ approaches 1 as $\alpha \to 2$ (Fig. 3). Additional validation using the Price network with $N = 10^4$ nodes and $M = 10^5$ edges, where $p(k) \propto k^{-\alpha}$, confirms that PA achieves higher AUC-ROC than the lower-bound and reaches near-maximal AUC-ROC scores for $\alpha \approx 2$.

We can also compute the AUC score using the Mann-Whitney U statistic (Fig. 4). Let us take a graph $G$ with the set of nodes given by $\mathcal{V}$. We sample nodes $i, j$ with degrees $k_i^+, k_j^+$ forming the positive set of edges from $p_{pos}$, and nodes $m, n$ with degrees $k_m^-, k_n^-$ forming the negative set of edges from $p_{neg}$. Then $P(k_i^+k_j^+ > k_m^-, k_n^-) \; \forall i, j, k, m \in \mathcal{V}$ is the AUC score and is given by $\frac{U}{n_1 n_2}$ where U is the Mann-Whitney U statistic and $n_1$, $n_2$ are sizes of the positive and negative edge sets respectively (Mason & Graham, 2002). We sample the random variables $k_i^+, k_j^+$ using the "Power_Law" function from the powerlaw package (Alstott et al., 2014) with degree exponent $\alpha - 1$ since $p_{pos}(k) \sim k^{-(\alpha-1)}$. Similarly, we sample $k_m^-, k_n^-$ with degree exponent $\alpha$ since $p_{neg}(k) \sim k^{-\alpha}$. Fig. 4 aligns with our findings in Fig. 3B, i.e., PA reaches near maximal AUC-ROC scores as $\alpha \to 2$.

### 4.4 Analysis of large-scale networks

To test whether the degree bias persists in large-scale real-world networks, we analyzed two large-scale citation networks, i.e., the Science of Science (SciSci) citation network (Lin et al., 2023), which represents citations between more than 95M publications across all sciences, and the USPTO citation network (Patent & Office, 2023), consisting of more than 7M patents in the US.

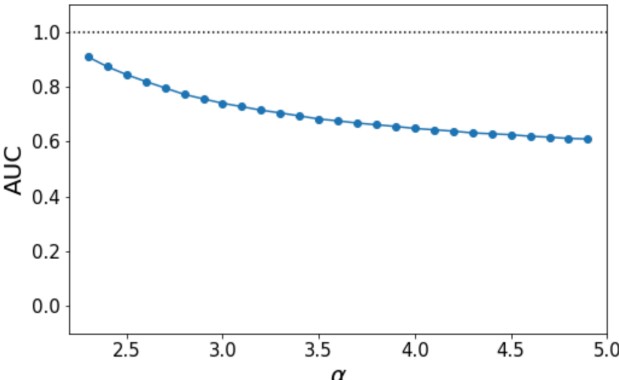

Figure 4: The influence of degree heterogeneity on the performance of the preferential attachment (PA) link prediction model in graphs with a power law degree distribution. The degree heterogeneity is governed by the power law exponent $\alpha$. As $\alpha$ increases, the heterogeneity decreases. PA reaches near maximal AUC-ROC scores (1) as $\alpha \to 2$. For each $\alpha$, we generate 20 batches, each with 5000 samples of $k_i^+, k_j^+, k_m^-, k_j^-$. The dots indicate the average AUC score obtained via the Mann-Whitney U statistic. Standard mean errors are smaller than the dots.

We followed the same procedure as our main analysis to test for degree bias, measuring the AUC-ROC score of the preferential attachment model. The results strongly supported our theoretical predictions: PA achieved an AUC-ROC score of 0.9452 for SciSci and 0.881 for USPTO, closely matching our theoretical predictions of 0.9479 and 0.918 respectively. With the degree-corrected benchmark, the AUC-ROC scores for PA decreases for both graphs, e.g., 0.5018 for SciSci and 0.4818 for the USPTO.

While we do not run other link prediction methods on these networks due to computational constraints, these results provide strong evidence that our findings about degree bias are not limited to smaller networks but represent a fundamental characteristic of the standard link prediction benchmark.

4.5 Parameter sensitivity in the analysis of performance alignment with the recommendation task

The vertex-centric max precision recall at $C$ (VCMPR@C) metric Menand & Seshadhri (2024) is a metric computed based on the precision and recall of the recommendations for each node. This metric is proposed for evaluating link prediction methods in recommendation settings. The VCMPR@C for a node $i$ and recommended node set $\mathcal{V}_i$ is defined as

$$\text{VCMPR@C for node } i = \frac{\sum_{j \in \mathcal{V}_i} Y_{ij}}{\max(C, m_i)}, \tag{13}$$

where $Y_{ij}$ is the indicator function of node $j$ is connected with $i$ in the test data ($Y_{ij} = 1$), and otherwise $Y_{ij} = 0$. Variable $m_i$ is the number of true connections in the test data, i.e., $m_i = \sum_j Y_{ij}$. We compute the average VCMPR@C for all nodes as the performance of the link prediction method for the graph.

We compute the similarity of two rankings with rank-biased overlap (RBO) Webber et al. (2010). RBO assesses the similarity of two rankings by examining the overlap of top-performing methods. Define $U_{k,1}$ as the set of methods ranked in the top $k$ positions in ranking 1, and $U_{k,2}$ similarly for ranking 2. Then, RBO computes a weighted average of the

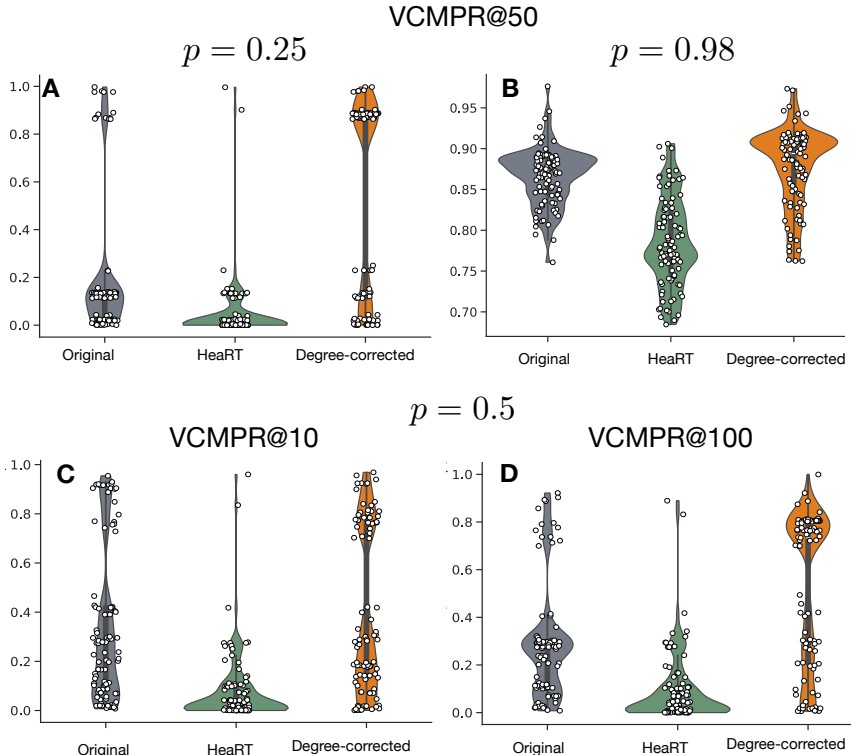

Figure 5: RBO for different $p$ values and different numbers $C$ of recommendations.

similarity of the top $k$ methods by

$$\text{RBO}(S, T, p) := (1 - p) \sum_{k=1}^{\infty} p^{k-1} \frac{|U_{k,1} \cap U_{k,2}|}{k}, \tag{14}$$

where $p$ controls the importance of the top performer, with a smaller $p$ value placing more weight on the top performer. We use $p = 0.5$ for the results in the main text. We find consistent results across different $p$ values (Fig. 5A and B). Additionally, we find consistent results for a different number of recommendations $C$ (Fig. 5C and D).

4.6  Evaluation of the link prediction performance using Hits@K

Hits@K is another common metric used to evaluate link prediction methods, alongside AUC-ROC. We investigated whether degree bias affects Hits@K scores and if our bias correction improves the correlation between Hits@K and actual link prediction performance. To compute Hits@K, we first ranked all test data edges by their predicted scores in descending order. We then counted the number of positive edges among the top K edges. This count was normalized by the maximum possible value (K) to indicate how close the method came to perfect prediction.

Our analysis revealed that the Hits@K scores for PA are consistently high across most networks, with only a few exceptions in networks with very low degree heterogeneity (Fig. 6A–D). The supriously high performance of PA remains consistent regardless of the K value used.

When we applied the same analysis to the degree-corrected benchmark, we found that the Hits@K scores for PA span the range between 0 to 1.0 (Fig. 6E–H). This indicates that the degree correction effectively mitigates spurious results and prevents inflation of performance metrics due to degree bias.

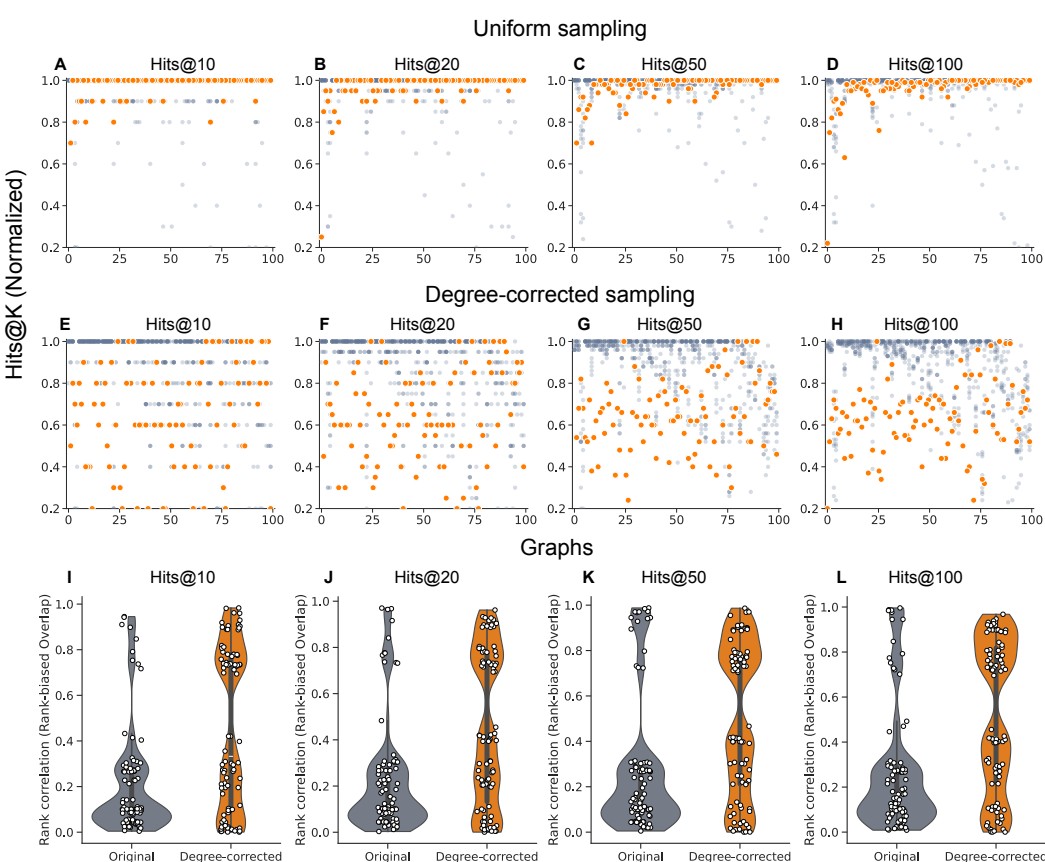

Figure 6: Link prediction performance using Hits@K. A–D Hits@K score for the PA model for the standard benchmark. E–H Hits@K score for the PA model for the degree-corrected benchmark. I–L The RBO score between the Hits@K scores and the the recommendation performance measured by VCMPR@50.

Table 3: Performance of GNNs trained using the HeaRT benchmark for networks with different mixing rates $\mu$.

| Mixing rate | GAT | GCN | GIN | GraphSAGE |
|---|---|---|---|---|
| 0.10 | 0.00668 | 0.04247 | 0.00603 | -0.00063 |
| 0.15 | 0.01600 | 0.01383 | 0.02049 | -0.00004 |
| 0.20 | 0.00951 | 0.01670 | 0.00955 | -0.00023 |

We then compared the Hits@K scores with the actual link prediction performance, measured by VCMPR@C, using RBO scores. Our results show that the RBO scores for the degree-corrected benchmark tend to be higher than those for the standard benchmark (Fig. 6I–L). This suggests that the degree-corrected benchmark provides a more accurate assessment of link prediction performance.

In summary, whether using AUC-ROC or ranking-based metrics like Hits@K, the underlying data used for evaluation is crucial. Our findings reveal a systemic problem: the data itself, when not properly corrected, has a bias that skews results in a way that is difficult to mitigate through metric selection alone. These results highlight the importance of addressing degree bias in link prediction evaluations, regardless of the metric used, and emphasize the need for careful evaluation methods in graph-based recommendation systems to ensure benchmark performance accurately reflects real-world performance.

### 4.7 Evaluation of the community detection performance using the normalized mutual information

Normalized Mutual Information (NMI) is a standard metric for assessing community detection methods Lancichinetti & Fortunato (2009); Fortunato & Hric (2016). NMI quantifies the similarity between actual and predicted community assignments, where a score of zero indicates no similarity. We note that NMI has a bias favoring partitions with small communities Gates et al. (2019), and thus, we used the element-centric similarity that does not have this bias in our main experiment. We note that NMI has a bias favoring partitions with small communities Gates et al. (2019), and thus, we used the element-centric similarity that does not have this bias in our main experiment. Nevertheless, we include the results for NMI in Fig. 7 for comparison. As with the element-centric similarity, our results show that the degree-corrected GNNs perform on par or better than the original GNNs.

### 4.8 Sensitivity to the choice of the LFR benchmark parameters

We tested the robustness of the results by using different parameter values for the LFR benchmark. First, we confirmed the consistent results when varying the average degree $\langle k \rangle$ from 25 to 50 (Fig. 8), or the maximum community size and degree from 1000 to 500 (Fig. 9).

## 5 GNNs trained with the HeaRT benchmark

We compared our degree-corrected benchmark with the distance-aware HeaRT benchmark (Li et al., 2024). Due to the computational expense of the negative sampling process, we were only able to analyze networks with small mixing rates ($\mu = 0.1$–0.20) where communities are well-separated.

The results when training GNNs on the HeaRT benchmark showed notably lower performance compared to both the standard and degree-corrected benchmarks:

For comparison, the performance of GNNs trained on both the standard and degree-corrected benchmarks exceeded 0.2 within this range of mixing rates.

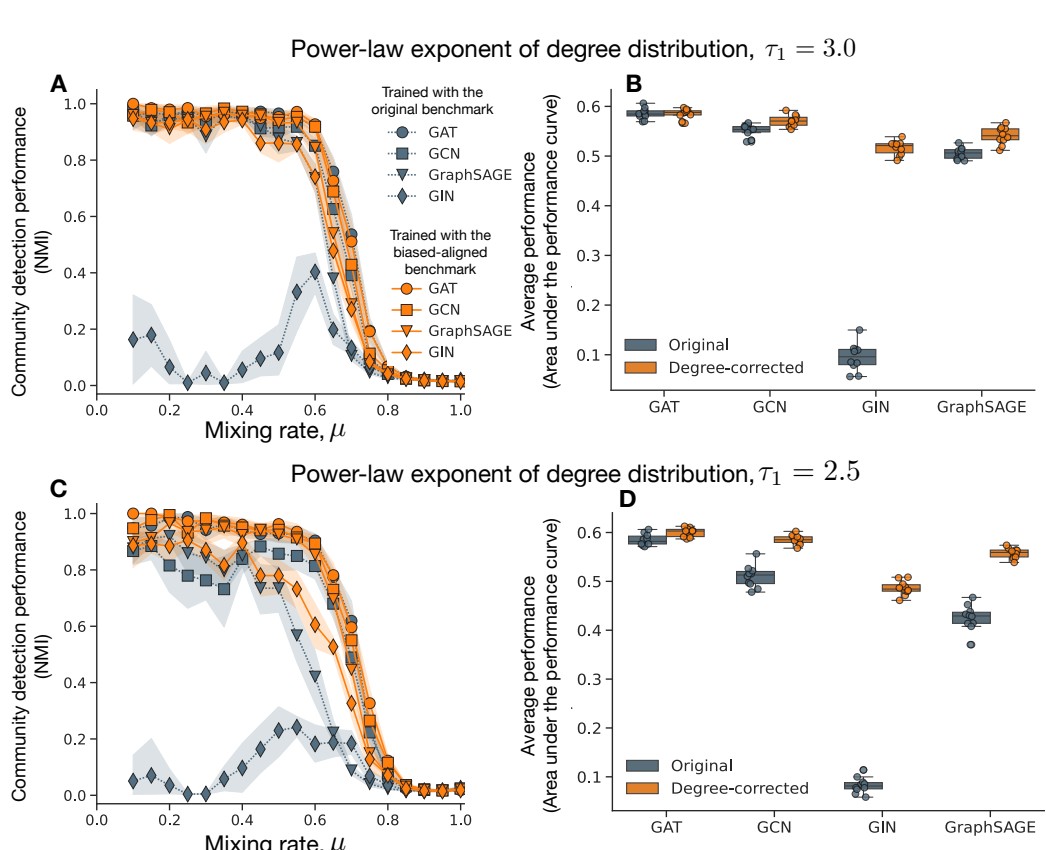

Figure 7: Performance of the GNNs on the LFR benchmark measured by NMI.

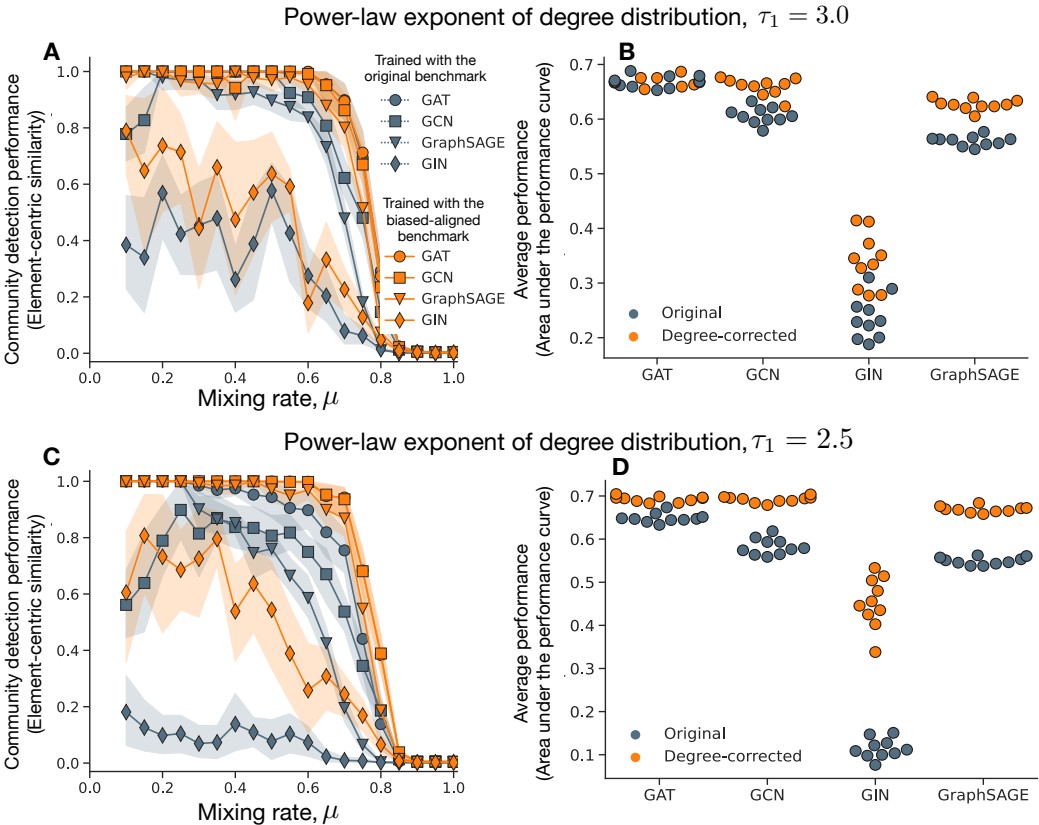

Figure 8: Performance of the GNNs on the LFR benchmark measured by NMI when varying the average degree $\langle k \rangle$ from 25 to 50.

## 6   Correlation between AUC-ROC and other network statistics

Figure 10A shows the correlation between AUC-ROC and other network statistics. We observed that the AUC-ROC of PA is strongly correlated with the degree heterogeneity in terms of the variance of the log-normal distribution of node degrees, more than other network statistics.

Figure 10B shows the correlation between AUC-ROC and the models that outperform PA on the standard benchmark. We observed that these models exhibit substantially weaker correlation between their AUC-ROC and degree heterogeneity, suggesting that their performance is not strongly tied to degree heterogeneity.

## 7   Reproducibility

### 7.1   Source data and code

The source data, code, and workflow for our experiments are available on GitHub and FigShare. The URLs are omitted in accordance with NeurIPS anonymity guidelines; however, we provide the data and code in the supplementary materials.

### 7.2   Snakemake workflow

We ensure the reproducibility of our experiments by using Snakemake Köster & Rahmann (2012), which allows automatic workflow execution from the preprocessing to the generation

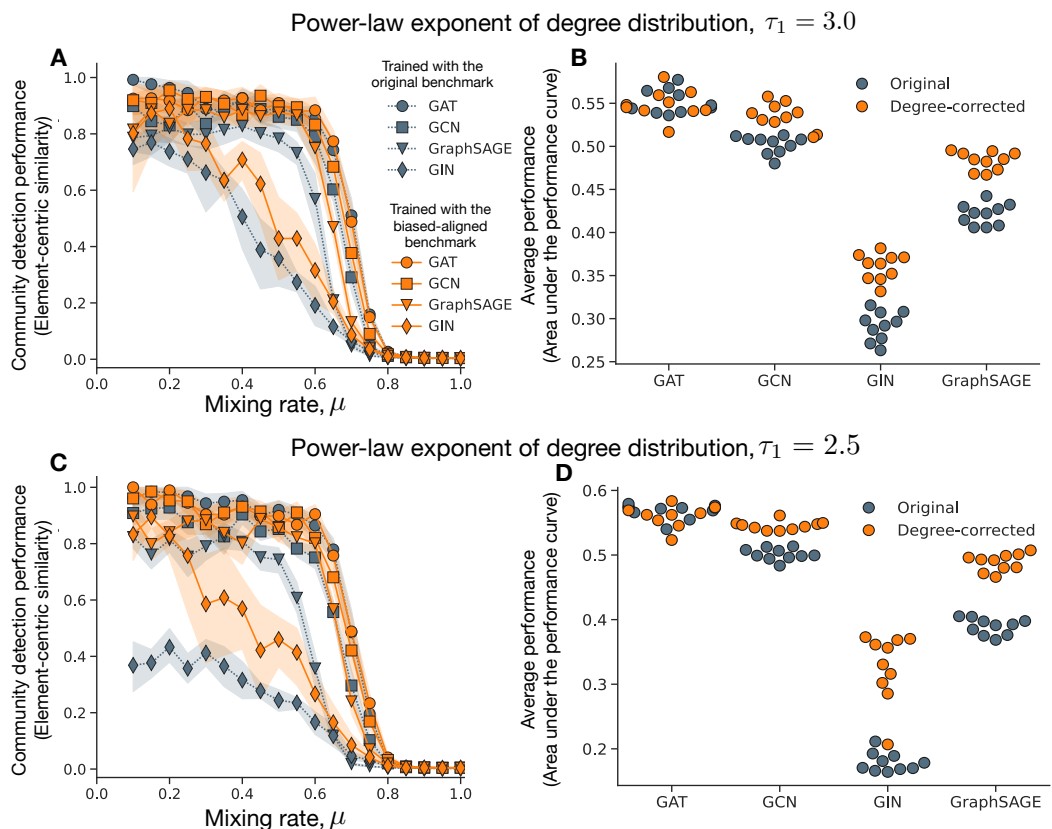

Figure 9: Performance of the GNNs on the LFR benchmark measured by NMI when varying the maximum community size and degree from 1000 to 500.

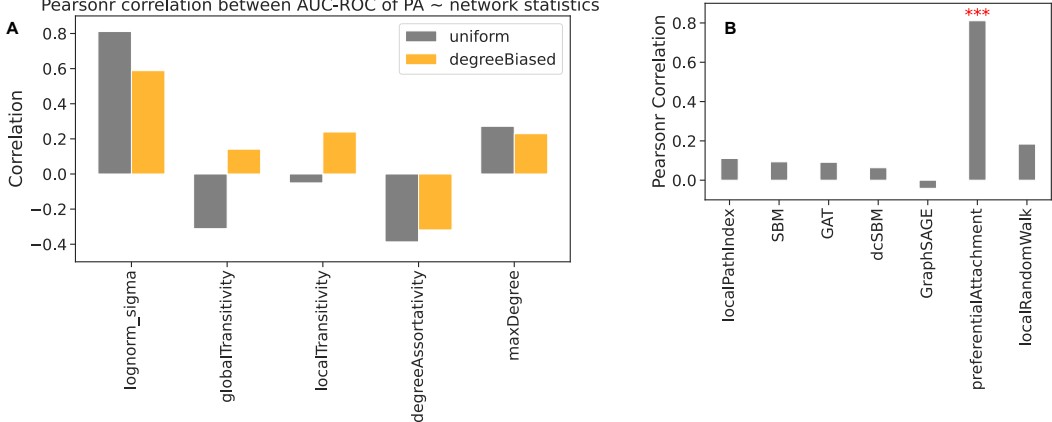

Figure 10: Correlation between AUC-ROC and other network statistics.

of the plots. With the Snakemake workflow, the user can reproduce all results by running the following command in the terminal:

```
snakemake --cores <number of cores> all
```

The workflow requires Python 3.11 or later, and all required Python packages are listed in the "environment.yaml" file in the repository.

### 7.3 Execution time and hardware requirements

We run the workflow on a server with 64 Intel(R) Xeon(R) Gold 5218 CPUs equipped with 64 cores, 1T RAM, and four NVIDIA GPUs with 48 GB memory, sufficient to complete the workflow in one week. The execution time of the workflow for the community detection task is 4 days, and that for the link prediction task is 10 days. The workflow can be executed with fewer resources by reducing the number of cores. The minimum computer requirements to run the workflow are as follows:

- 64 GB RAM
- 16 GB GPU memory
- 8 core CPU
- 300 GB space

Table 4: Network data tested in this study. We consider social, technological, information, biological, and transportation (spatial) networks. For simplicity in our analysis, we consider these networks to be unweighted, undirected, and without self-loops. Variance referes to the variance of the node degrees. Assortativity refers to the degree assortativity, and Heterogeneity refers to the degree heterogeneity computed by Jacob et al. (2017).

| Network | Nodes | Edges | Max. Degree | Variance | Assortativity | Heterogeneity |
|---|---|---|---|---|---|---|
| Political books | 105 | 441 | 25 | 29.69 | -0.128 | 0.43 |
| College football | 115 | 613 | 12 | 0.78 | 0.162 | 0.20 |
| High school 2011 | 126 | 1709 | 55 | 153.71 | 0.083 | 0.62 |
| Food web bay wet | 128 | 2075 | 110 | 249.17 | -0.112 | 0.62 |
| Food web bay dry | 128 | 2106 | 110 | 249.85 | -0.104 | 0.63 |
| Radoslaw email | 167 | 3250 | 139 | 993.84 | -0.295 | 0.62 |
| Highschool 2012 | 180 | 2220 | 56 | 120.37 | 0.046 | 0.51 |
| Little Rock Lake | 183 | 2434 | 105 | 433.29 | -0.266 | 0.54 |
| Jazz | 198 | 2742 | 100 | 303.12 | 0.020 | 0.55 |
| C. Elegans | 297 | 2148 | 134 | 167.56 | -0.163 | 0.38 |
| Network science | 379 | 914 | 34 | 15.42 | -0.082 | 0.23 |
| Dublin social | 410 | 2765 | 50 | 70.51 | 0.226 | 0.29 |
| Airport | 500 | 2980 | 145 | 499.03 | -0.268 | 0.35 |
| Caltech | 762 | 16651 | 248 | 1365.76 | -0.066 | 0.42 |
| Reed | 962 | 18812 | 313 | 1254.53 | 0.023 | 0.38 |
| Political blogs | 1222 | 16714 | 351 | 1474.67 | -0.221 | 0.34 |
| Haverford | 1446 | 59589 | 375 | 3687.70 | 0.067 | 0.41 |
| Simmons | 1510 | 32984 | 300 | 1288.53 | -0.062 | 0.32 |
| Swarthmore | 1657 | 61049 | 577 | 3472.20 | 0.061 | 0.37 |
| Petster | 1788 | 12476 | 272 | 440.86 | -0.089 | 0.24 |
| UC Irvine | 1893 | 13835 | 255 | 599.57 | -0.188 | 0.24 |
| Yeast | 2224 | 6609 | 64 | 63.67 | -0.105 | 0.15 |
| Amherst | 2235 | 90954 | 467 | 4007.71 | 0.058 | 0.35 |
| Bowdoin | 2250 | 84386 | 670 | 3206.35 | 0.056 | 0.33 |
| Hamilton | 2312 | 96393 | 602 | 3940.69 | 0.031 | 0.34 |
| Adolescent health | 2539 | 10455 | 27 | 18.59 | 0.251 | 0.10 |
| Trinity | 2613 | 111996 | 404 | 3742.49 | 0.072 | 0.32 |
| USFCA | 2672 | 65244 | 405 | 2041.31 | 0.092 | 0.27 |
| Japanese book | 2698 | 7995 | 725 | 608.58 | -0.259 | 0.17 |
| Williams | 2788 | 112985 | 610 | 3901.94 | 0.040 | 0.32 |
| Open flights | 2905 | 15645 | 242 | 485.44 | 0.049 | 0.21 |
| Oberlin | 2920 | 89912 | 478 | 2838.11 | 0.050 | 0.28 |
| Wellesley | 2970 | 94899 | 746 | 3079.68 | 0.064 | 0.29 |
| Smith | 2970 | 97133 | 349 | 2432.51 | 0.044 | 0.28 |
| Vassar | 3068 | 119161 | 482 | 3453.23 | 0.101 | 0.30 |
| Middlebury | 3069 | 124607 | 473 | 3865.24 | 0.078 | 0.30 |

Table 5: (Continued) Network data tested in this study.

| Network | Nodes | Edges | Max. Degree | Variance | Assortativity | Heterogeneity |
|---|---|---|---|---|---|---|
| Pepperdine | 3440 | 152003 | 674 | 5695.91 | 0.055 | 0.31 |
| Colgate | 3482 | 155043 | 773 | 4009.14 | 0.067 | 0.29 |
| Santa | 3578 | 151747 | 1129 | 4933.35 | 0.071 | 0.29 |
| Wesleyan | 3591 | 138034 | 549 | 3548.92 | 0.095 | 0.28 |
| Mich | 3745 | 81901 | 419 | 1997.51 | 0.142 | 0.24 |
| Bitcoin alpha | 3775 | 14120 | 511 | 402.89 | -0.169 | 0.17 |
| Bucknell | 3824 | 158863 | 506 | 3498.69 | 0.094 | 0.27 |
| Brandeis | 3887 | 137561 | 1972 | 4646.98 | -0.026 | 0.27 |
| Howard | 4047 | 204850 | 1215 | 8506.41 | 0.058 | 0.32 |
| Rice | 4083 | 184826 | 581 | 5669.22 | 0.065 | 0.29 |
| GR-QC 1993-2003 | 4158 | 13422 | 81 | 74.41 | 0.639 | 0.12 |
| Tennis | 4338 | 81865 | 451 | 4573.31 | 0.003 | 0.26 |
| Rochester | 4561 | 161403 | 1224 | 3632.47 | 0.025 | 0.25 |
| Lehigh | 5073 | 198346 | 973 | 4073.33 | 0.035 | 0.24 |
| JohnsHopkins | 5157 | 186572 | 886 | 4761.94 | 0.080 | 0.25 |
| HT09 | 5352 | 18481 | 1287 | 1333.44 | -0.431 | 0.14 |
| Wake | 5366 | 279186 | 1341 | 7469.92 | 0.071 | 0.27 |
| Hep-Th 1995-99 | 5835 | 13815 | 50 | 20.77 | 0.185 | 0.08 |
| Bitcoin OTC | 5875 | 21489 | 795 | 531.22 | -0.165 | 0.15 |
| Reactome | 5973 | 145778 | 855 | 4612.48 | 0.241 | 0.21 |
| Jung | 6120 | 50290 | 5655 | 16029.25 | -0.233 | 0.16 |
| Gnutella Aug 08 2002 | 6299 | 20776 | 97 | 72.95 | 0.036 | 0.11 |
| American | 6370 | 217654 | 930 | 3847.11 | 0.066 | 0.22 |
| MIT | 6402 | 251230 | 708 | 6241.81 | 0.120 | 0.24 |
| JDK | 6434 | 53658 | 5923 | 16112.86 | -0.223 | 0.16 |
| William | 6472 | 266378 | 1124 | 5164.22 | 0.052 | 0.23 |
| U Chicago | 6561 | 208088 | 1624 | 4093.91 | 0.018 | 0.22 |
| Princeton | 6575 | 293307 | 628 | 6164.10 | 0.091 | 0.24 |
| Carnegie | 6621 | 249959 | 840 | 5674.47 | 0.122 | 0.24 |
| Tufts | 6672 | 249722 | 827 | 4525.50 | 0.118 | 0.22 |
| UC | 6810 | 155320 | 660 | 2297.32 | 0.125 | 0.19 |
| Wikipedia elections | 7066 | 100736 | 1065 | 3332.59 | -0.083 | 0.21 |
| English book | 7377 | 44205 | 2568 | 3699.80 | -0.237 | 0.16 |
| Gnutella Aug 09 2002 | 8104 | 26008 | 102 | 66.74 | 0.033 | 0.09 |

Table 6: (Continued) Network data tested in this study.

| Network | Nodes | Edges | Max. Degree | Variance | Assortativity | Heterogeneity |
|---|---|---|---|---|---|---|
| French book | 8308 | 23832 | 1891 | 1217.86 | -0.233 | 0.12 |
| Hep-Th 1993-2003 | 8638 | 24806 | 65 | 41.61 | 0.239 | 0.08 |
| Gnutella Aug 06 2002 | 8717 | 31525 | 115 | 51.87 | 0.052 | 0.09 |
| Gnutella Aug 05 2002 | 8842 | 31837 | 88 | 54.66 | 0.015 | 0.09 |
| PGP | 10680 | 24316 | 205 | 65.24 | 0.238 | 0.09 |
| Gnutella Aug 04 2002 | 10876 | 39994 | 103 | 48.65 | -0.013 | 0.08 |
| Hep-Ph 1993-2003 | 11204 | 117619 | 491 | 2307.04 | 0.630 | 0.16 |
| Spanish book 1 | 11558 | 43050 | 2986 | 3353.23 | -0.282 | 0.12 |
| DBLP citations | 12495 | 49563 | 709 | 284.34 | -0.046 | 0.10 |
| Spanish book 2 | 12643 | 55019 | 5169 | 6953.72 | -0.290 | 0.11 |
| Cond-Mat 1995-99 | 13861 | 44619 | 107 | 45.70 | 0.157 | 0.07 |
| Astrophysics 1 | 14845 | 119652 | 360 | 472.92 | 0.228 | 0.11 |
| Astrophysics 2 | 17903 | 196972 | 504 | 961.58 | 0.201 | 0.11 |
| Cond-Mat 1993-2003 | 21363 | 91286 | 279 | 119.00 | 0.125 | 0.07 |
| Gnutella Aug 25 2002 | 22663 | 54693 | 66 | 28.58 | -0.173 | 0.04 |
| Internet | 22963 | 48436 | 2390 | 1085.20 | -0.198 | 0.08 |
| Thesaurus | 23132 | 297094 | 1062 | 1993.31 | -0.048 | 0.12 |
| Cora | 23166 | 89157 | 377 | 123.05 | -0.055 | 0.07 |
| AS Caida | 26475 | 53381 | 2628 | 1113.83 | -0.195 | 0.08 |
| Gnutella Aug 24 2002 | 26498 | 65359 | 355 | 35.03 | -0.008 | 0.04 |
| ogbl-collab | 232865 | 961883 | 382 | 178.857773 | 0.269877 | 1.134311 |
| ogbl-ddi | 4267 | 1067911 | 2234 | 176801.815426 | 0.037832 | 0.730724 |
| ogbl-biokg-protein | 11034 | 884042 | 2551 | 62766.451816 | -0.027401 | 1.112268 |
| ogbl-biokg-drug | 7313 | 137027 | 652 | 15456.627212 | 0.079466 | 1.577354 |
| ogbl-biokg-function | 44635 | 1180424 | 17690 | 61471.922776 | -0.128156 | 1.770447 |

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
