# OpenReview forum: "Implicit degree bias in the link prediction task"
_ICLR.cc/2025/Conference — Submitted to ICLR 2025_

### Official Review · Reviewer_mfmk · 2024-11-01

**Soundness:** 3
**Presentation:** 3
**Contribution:** 3
**Rating:** 6
**Confidence:** 4

**Summary:**

The paper illustrates the presence of an implicit bias toward high-degree nodes in the usual link prediction benchmark for graphs, potentially favoring prediction methods based on node degree (e.g. preferential attachment).

Then, authors propose a straightforward yet effective idea to address this issue by sampling negative edges with the same biased distribution as positive ones. This novel benchmark is shown to align better with link recommendation tasks. Moreover, the degree-corrected benchmark allows for training unsupervised GNN models that better capture graph structure.

**Strengths:**

The paper provides a detailed analysis of the standard benchmark for link prediction, revealing a bias in positive edge sampling that can lead to misleading experimental outcomes. The proposed solution is very straightforward, not very original, but effective across the experiments. While the exploration of biases in link prediction is not a new concept, the paper's contribution to improving current benchmarks for more reliable and fair results represents a valuable advancement. The clarity of the paper is generally good, particularly in the introductory sections, while there is room for improvement in the description of experiments. I have also appreciated the theoretical analyses and the amount of supplementary results reported, which ensure the overall quality of the work.

**Weaknesses:**

I have noticed two main weaknesses. First, aspects of the exposition lack clarity. Specifically, in Section 3.2, where authors explore the alignment with recommendation tasks, the task description is somewhat vague, and the section would benefit from a substantial revision to improve coherence and integration with the rest of the paper. Second, the impact of biased negative sampling during GNN training seems underexplored in the experiments. In fact, Section 3.3 demonstrates a positive effect when graph models are trained with bias-corrected sampling, suggesting that improved community learning is due to the reduced overfitting to node degrees. However, other factors could also contribute to this improvement. To clarify the sources of enhancement in community detection with degree-corrected sampling, authors should include additional experiments or more detailed discussions. I elaborate further on these issues and other minor concerns in the question box below.

**Questions:**

(1) Section 3.2 provides a vague and unclear intuition about recommendation tasks. I strongly recommend revising this section, possibly incorporating some of the information already included in the appendix. Specifically, authors should include a clear definition of the recommendation task objective, metrics used for evaluation, and an explanation of why alignment between link prediction and recommendation is important. For instance, I wonder why not directly using link recommendation instead of a new link prediction benchmark if already the first one is less biased?
Additionally, as a minor comment, the results presented in Figure 2B are not easy to interpret. In particular, I do not understand why using a different color for grey and orange distributions and which part of the graph highlights the alignment with recommendation tasks. Improving the clarity with a more detailed caption and/or adding labels to the picture is important for understanding the results more deeply. Moreover, the figure should be clearly referenced in the corresponding paragraph where it is discussed.

(2) Section 3.3 demonstrates notable improvements in community learning when degree-corrected sampling is applied during GNN training, suggesting that node degree overfitting is the primary factor in performance loss for community detection. However, other factors may also contribute to this improvement. For example, as mentioned in the final discussion, correcting for degree bias also reduces distance bias, making link prediction more challenging by removing easily distinguishable negative examples and thereby promoting more robust structure learning. Authors should explore more in depth this possibility, proving evidence that reduction in node degree overfitting, beyond the elimination of other possible biases, is the main reason for enhanced community learning. At this purpose, adding in Figure 3 the results with a "distance-unbiased" benchmark can be an important comparison, or alternatively, explaining why the use of the degree-corrected benchmark should be preferred to reduce distance bias (e.g., computationally faster?).

(3) Related to the previous question, Section 3.3 mentions that GNNs are trained using either the original and degree-corrected link prediction benchmarks and tested on community detection. Have you also evaluated how link prediction performance is affected —across both the standard and degree-corrected benchmarks— when training is conducted without the bias?

(4) Figures 3B and 3D show an overall improvement in learning community structure when using the degree-corrected benchmark. However, the performance gain with GAT appears minimal. Do you have any insights or hypotheses as to why this might be the case?

(5) Given that in the degree-corrected benchmark, negative sampling follows the same degree bias as positive edges, can you provide any theoretical guarantee, empirical evidence, or link to other works, showing that the pairs involving low-degree nodes (mostly neglected during sampling) have a minor impact on prediction accuracy?

---

> ### Author Response · Authors · 2024-11-26
>
> > Questions:
> > (1) Section 3.2 provides a vague and unclear intuition about recommendation tasks. I strongly recommend revising this section, possibly incorporating some of the information already included in the appendix. Specifically, authors should include a clear definition of the recommendation task objective, metrics used for evaluation, and an explanation of why alignment between link prediction and recommendation is important. For instance, I wonder why not directly using link recommendation instead of a new link prediction benchmark if already the first one is less biased?
>
> We appreciate referee for making this excellent point.
> We have revised Section 3.2 to make the connections between link prediction and recommendation tasks more explicit.
>
> Link prediction serves as a computationally efficient proxy for evaluating and training recommendation systems. While the ultimate goal is often recommendation, directly optimizing recommendation metrics such as Hits@K requires ranking all possible node pairs for each node, which is computationally infeasible for large networks (${\cal O}(N^2)$, where $N$ is the number of nodes). In contrast, link prediction evaluates on a fixed set of candidate pairs, making it ${\cal O}(M)$ (where $M$ is the number of edges) and thus practical for both evaluation and training.
>
> This computational advantage has made link prediction benchmarks a de facto standard for developing and training recommendation models.
> However, this practice is only valid if link prediction performance correlates with recommendation performance.
> Our results show that degree bias in standard link prediction benchmark diminish this correlation.
> The degree-corrected benchmark improves this correlation (Fig 2B), providing a more reliable yet still computationally efficient proxy for recommendation performance.
>
> In the revised manuscript, we have clarified the motivation of this experiment and that in the recommendation task, a method must rank all potential connections for each node without a predefined candidate set. This differs fundamentally from link prediction where we evaluate on a fixed set of node pairs.
>
> We believe that the revised section now provides a more thorough explanation of the relationship between link prediction and recommendation tasks.
>
> > Additionally, as a minor comment, the results presented in Figure 2B are not easy to interpret. In particular, I do not understand why using a different color for grey and orange distributions and which part of the graph highlights the alignment with recommendation tasks. Improving the clarity with a more detailed caption and/or adding labels to the picture is important for understanding the results more deeply. Moreover, the figure should be clearly referenced in the corresponding paragraph where it is discussed.
>
> We admit that the explanation could be more clear by enriching the figure with more annotations, and thus in the new figure 2B, we have added a legend that explains that individual circles represent the similarity of method ranking for individual graphs.
> In addition to the mere ranking similarities, we also plotted the violin plot because multiple circles overlap and it is not easy to see the density of the data points.
> Thus, we also included the violin plot to provide more information about the distribution of the data.
> Regarding the color choices, we have used different colors to indicate that the samples come from different benchmarks.

---

> ### Author Response · Authors · 2024-11-26
>
> > (2) Section 3.3 demonstrates notable improvements in community learning when degree-corrected sampling is applied during GNN training, suggesting that node degree overfitting is the primary factor in performance loss for community detection. However, other factors may also contribute to this improvement. For example, as mentioned in the final discussion, correcting for degree bias also reduces distance bias, making link prediction more challenging by removing easily distinguishable negative examples and thereby promoting more robust structure learning. Authors should explore more in depth this possibility, proving evidence that reduction in node degree overfitting, beyond the elimination of other possible biases, is the main reason for enhanced community learning. At this purpose, adding in Figure 3 the results with a "distance-unbiased" benchmark can be an important comparison, or alternatively, explaining why the use of the degree-corrected benchmark should be preferred to reduce distance bias (e.g., computationally faster?).
>
> The referee makes a very good point. While we attempted to include a distance-unbiased benchmark (HeaRT; Li et al., NeurIPS 2024), we find that it is computationally infeasible to run the experiments within the revision cycle. This is because the negative sampling of HeaRT involves the computation of PageRank, which is computationally expensive, and GNN training requires substantially more negative samples than evaluation.
> While we are not able to provide the full performance spectrum, we could manage to run the experiments on networks with small mixing rates (0.1-0.20) where communities are well-separated. Results when trained on the HeaRT benchmark:
>
> | Mixing rate |   GAT |   GCN |   GIN |   GraphSAGE |
> |-----:|--------------------:|--------------------:|--------------------:|--------------------------:|
> | 0.1  |             0.00668 |             0.04247 |             0.00603 |                  -0.00063 |
> | 0.15 |             0.016   |             0.01383 |             0.02049 |                  -4e-05   |
> | 0.2  |             0.00951 |             0.0167  |             0.00955 |                  -0.00023 |
>
>
> These accuracies are notably lower than models trained on original and degree-corrected benchmarks.
> For reference, the performance of GNNs trained on the standard and degree-corrected benchmarks are all above 0.2 within this range of mixing rates.
>
> While we avoid definitive claims, this may be because HeaRT may excessively suppress distance information, which is an important feature for link prediction.
> Nodes tend to connect more to nearby nodes than distant ones in LFR graphs, and links are more likely to appear within communities than between them.
> Training on a distance-unbiased benchmark effectively forces GNNs to treat all node pairs equally regardless of their distance, thus suppressing this crucial community signal.
> Consequently, the model struggles to learn that nodes within the same community have higher connectivity, making it harder to identify coherent community structures.
>
> > (3) Related to the previous question, Section 3.3 mentions that GNNs are trained using either the original and degree-corrected link prediction benchmarks and tested on community detection. Have you also evaluated how link prediction performance is affected —across both the standard and degree-corrected benchmarks— when training is conducted without the bias?
>
> Yes, we have evaluated the link prediction performance across both benchmarks.
> We observe that GNNs trained on the degree-corrected benchmark show slightly improved average rankings compared to their performance on the original benchmark
> However, we consider that this improvement is a natural consequence of models performing better on evaluation metrics that match their training objective.
> Namely, GNNs trained on the degree-corrected benchmark perform better on the degree-corrected benchmark, and vice versa.
> This has led us to an independent validation based on community detection, where the training objective is related but different from the evaluation metric.

---

> ### Author Response · Authors · 2024-11-26
>
> > (4) Figures 3B and 3D show an overall improvement in learning community structure when using the degree-corrected benchmark. However, the performance gain with GAT appears minimal. Do you have any insights or hypotheses as to why this might be the case?
>
> Thank you for the question. The minimal improvement seen with GAT can be explained by its already strong baseline performance. When we tested Infomap, which is considered the best performing community detection method for the LFR benchmark [1,2], it achieved average performance scores of 0.42 ($\tau_1 = 2.5$) and 0.35 ($\tau_1 = 3$) on our LFR graphs. These scores are actually lower than GAT's performance without degree-correction (Fig. 3B).
> Since GAT is already performing exceptionally well, we believe there is simply less room for improvement through degree-correction.
>
> [1] A. Lancichinetti, S. Fortunato, and F. Radicchi, "Benchmark graphs for testing community detection algorithms," Phys. Rev. E, vol. 78, no. 4, p. 046110, 2008.
> [2] S. Kojaku, F. Radicchi, Y.Y. Ahn, and S. Fortunato. "Network community detection via neural embeddings." Nature Communications 15, (2024): 9446.
>
> > (5) Given that in the degree-corrected benchmark, negative sampling follows the same degree bias as positive edges, can you provide any theoretical guarantee, empirical evidence, or link to other works, showing that the pairs involving low-degree nodes (mostly neglected during sampling) have a minor impact on prediction accuracy?
>
> This is an excellent question about potential undersampling of low-degree nodes in our degree-corrected benchmark.
> A key to consider this question is that prediction performance is measured through *comparisons between positive and negative edges*, not between positive edges and not between negative edges.
> This means that sampling more negative edges involving low-degree nodes while using the same positive edges does not make the performance metric more representative of low-degree nodes.
> Instead, this creates an adversarial effect.
> If low-degree nodes are sampled as negative edges more frequently, this creates more bias by overrepresenting comparisons between high-degree positives and low-degree negatives.
> In the revised manuscript, we have provided theoretical and empirical evidence that this is indeed the case (Section 3 in the SI).
> We also demonstrated that the degree-corrected benchmark produces more balanced comparisons by sampling the negative edges with the same distribution as positive edges.

---

> > ### Comment · Reviewer_mfmk · 2024-11-26
> >
> > Thank you to the authors for their detailed and thorough responses. For now, I will maintain my ratings but reserve the possibility of updating them during discussions with the other reviewers.

---

### Official Review · Reviewer_Ttnj · 2024-11-02

**Soundness:** 3
**Presentation:** 3
**Contribution:** 3
**Rating:** 8
**Confidence:** 3

**Summary:**

This paper critically examines the standard link prediction benchmark in graph machine learning, revealing an inherent degree bias that favors methods overly dependent on node degree. The authors propose a degree-corrected link prediction task to address this bias, demonstrating its improved alignment with real-world recommendation tasks and its ability to train graph machine learning models more effectively.

**Strengths:**

1. The paper is well-structured and clearly articulates the problem, proposed solution, and implications.
2. The figures and their captions do very well to illustrate the problem and approach
3. The findings have broad implications for graph machine learning, and could potentially influence future benchmark designs and model evaluations.

**Weaknesses:**

1. The paper focuses primarily on undirected, unweighted graphs. It's unclear how the findings generalize to other graph types (e.g., directed, weighted). (see Q1)
2. The practical implications for existing graph machine learning models and their reported performances are not extensively discussed.

**Questions:**

1. Any intuition on how the degree-corrected benchmark would perform on directed or weighted graphs?
2. Elaborating on how the degree-corrected benchmark might be integrated into existing graph machine learning frameworks and evaluation pipelines would benefit the paper.

---

> ### Author Response · Authors · 2024-11-26
>
> We thank the referee for this insightful questions! For directed graphs, the degree-correction would need to consider both in-degree and out-degree distributions separately. The sampling process would need to be modified to maintain the same bias in both distributions when sampling negative edges. For weighted graphs, our degree-correction principle could be extended by considering weighted degrees, though this would require careful consideration of how edge weights interact with the sampling process. We have added this discussion in the Discussion section of the revised manuscript.
>
> For Question 2:
> The degree-corrected benchmark can be readily integrated into existing frameworks as it only modifies the negative edge sampling procedure. This was demonstrated in the task of community detection, where we generate the training edges by the degree-corrected benchmarks.

---

> > ### Comment · Reviewer_Ttnj · 2024-11-27
> >
> > Noted. I would like to thank the authors for their response.

---

### Official Review · Reviewer_fa1W · 2024-11-03

**Soundness:** 3
**Presentation:** 3
**Contribution:** 2
**Rating:** 5
**Confidence:** 5

**Summary:**

Real-world graphs are sparse and have a heavy-tailed degree distribution. Thus, when measuring performance on link prediction, one should not randomly sample a bunch of edges (positive examples) and non-edges (negative examples). The paper presents a degree-corrected link prediction benchmark, where the non-edge samples have the same degree bias as edges.  The paper shows how the degree-corrected sampling learns better community structure.

**Strengths:**

The paper talks about an important problem and is well-written. The proposed solution is reasonable for graphs with no attributes.

**Weaknesses:**

The problem highlighted is well-known. For example, here are two recent works:

[1] Xie He et al. (2024). Link prediction accuracy on real-world networks under non-uniform missing-edge patterns. PLoS One, 18;19(7):e0306883. https://journals.plos.org/plosone/article?id=10.1371/journal.pone.0306883

[2] Ayan Chatterjee et al. (2023). Improving the generalizability of protein-ligand binding predictions with AI-Bind. Nature Communications, 14, Article number: 1989. https://doi.org/10.1038/s41467-023-37572-z

**Questions:**

Xie He et a. [1] showed that one must consider the data collection process and application domain for link prediction. Are you suggesting that the degree-corrected sampling benchmark should be used for all graphs regardless of the application domain or the data collection process?

[1] Xie He et al. (2024). Link prediction accuracy on real-world networks under non-uniform missing-edge patterns. PLoS One, 18;19(7):e0306883. https://journals.plos.org/plosone/article?id=10.1371/journal.pone.0306883

---

> ### Author Response · Authors · 2024-11-26
>
> We thank the reviewer for highlighting important previous work, which empirically demonstrated that different negative edge sampling methods significantly affect link prediction performance.
> Their results raise a crucial question: what underlying mechanisms cause these performance differences?
> This question has not been answered elsewhere in link prediction literature theoretically and systematically.
>
> Our work directly addresses this question by identifying degree heterogeneity as a fundamental factor driving these performance variations.
> While Xie He et al. provided valuable empirical observations across different sampling strategies, their study was limited to small networks (< 900 nodes) where degree heterogeneity effects are relatively mild, making it difficult to observe the full impact of degree heterogeneity that becomes pronounced in real-world scale-free networks with millions of nodes.
>
> We provide a more comprehensive analysis through (1) large-scale evaluation on 95 networks including those with extreme degree heterogeneity, (2) theoretical analysis showing how exactly degree heterogeneity affects AUC-ROC scores, and (3) theoretical proof that in networks with high degree heterogeneity, even naive degree-based methods can achieve near-optimal performance.
>
> These together advances our understanding beyond empirical observations to fundamental mechanisms that affect link prediction in real-world networks.

---

> > ### Comment · Reviewer_fa1W · 2024-11-27
> >
> > I appreciate the authors feedback. However, the message of the paper is better suited for the NeurIPS Datasets and Benchmarks track.

---

### Official Review · Reviewer_1S2G · 2024-11-03

**Soundness:** 3
**Presentation:** 3
**Contribution:** 2
**Rating:** 5
**Confidence:** 4

**Summary:**

This work highlights how the "standard" setup for link prediction task in graph machine learning suffers from a sampling degree bias. This bias, as shown, disproportionately affects nodes with high degrees, making node degree a dominant factor in determining link presence, irrespective of actual structural features. The authors validate their claims through empirical evaluation of a wide range of datasets and some theoretical analysis on networks with a log-normal degree distribution. They show that a naive method that just considers the degree of nodes to make a prediction performs really well in the old setup and incrementally proposes a modification to the existing setup to counter this sampling bias.

**Strengths:**

1. Originality: This work identifies and addresses an under-examined issue in graph machine learning benchmarks - implicit degree bias in sampling in the "standard" link prediction benchmark. By identifying and correcting this bias, the authors contribute a novel perspective on the evaluation methodologies that underpin graph-based tasks. This originality lies in highlighting how benchmark designs can inadvertently favor certain features, like node degree, thus affecting model evaluation in unintended ways. The proposed degree-corrected benchmark, although quite simple, offers an approach for mitigating this bias.

2. Quality: The paper provides a thorough analysis of degree bias, including both theoretical insights (in a specific setting) and empirical validation. The experiments span a range of datasets and link prediction methods, offering a well-rounded evaluation of the proposed benchmark's effectiveness.

3. Clarity: The paper generally communicates its objectives, findings, and implications well, especially in its discussion of degree bias and its impact on link prediction tasks. The premise is clear from the very start and is easy to read and follow, although some figures are quite hard to follow. The discussion section is well written and properly acknowledges pitfalls and potential scope in the future.

4. Significance: This work has some implications for the evaluations in this field. With the potential bias identified, newer methods that do not evaluate in any already special configurations of the link prediction benchmarks can adopt this.

**Weaknesses:**

Let me preface by appreciating the thorough limitations discussed in the Discussion section. Although they are acknowledged and defended to some extent, I still feel that some of the explanations are not convincing enough, and hence, you might find me mentioning a couple of them down below.

W1. The authors overgeneralize the impact of their proposed benchmark. While the described setup was the most followed one, in recent times, other works tend to adopt different sampling schemes and experimental setups [3] [4]. While this work touches on an important issue with the legacy benchmarks, I wonder how relevant they are currently.

W2. Despite highlighting degree bias, the paper fails to thoroughly compare with recent works that address similar benchmarking biases. Relevant works, such as those by [1] [2], which tackle degree-related bias, are absent. This lack of engagement with the broader literature makes the contribution feel isolated rather than a comprehensive advancement in benchmark design.

W3. Hyperparameter tuning for GNNs is missing, which is paramount while benchmarking multiple methods. While the authors mention standard settings, there is no indication that they optimized parameters for the degree-corrected benchmark. I find the results brittle as it is common knowledge that most of the GNN-based methods are very sensitive to the hyperparameters, which, when properly tuned, could result in a completely different set of results. Another weakness on the same line of reasoning is that while the authors test on 96 graphs, most of their discussions are centered around the insights from artificial datasets, rendering the depth of the dataset choices valuable only to some extent.

W4. The choice of only four GNNs among 26 total methods seems outdated, as classical link prediction methods are now less commonly used. Incorporating a more diverse set of GNN models would better reflect the benchmark’s utility in current graph machine learning applications. Graph Transformers and their variants, which are often the most commonly used ones in practice now, are completely missing from the evaluations.

[1] Mayra Russo, Sammy Fabian Sawischa, and Maria-Esther Vidal. 2024. Tracing the Impact of Bias in Link Prediction. In Proceedings of the 39th ACM/SIGAPP Symposium on Applied Computing (SAC '24).

[2] Y. Wang and T. Derr, "Degree-Related Bias in Link Prediction," 2022 IEEE International Conference on Data Mining Workshops (ICDMW)

[3] Wang, Y., Hu, X., Gan, Q., Huang, X., Qiu, X., & Wipf, D. (2024). Efficient Link Prediction via GNN Layers Induced by Negative Sampling. IEEE Transactions on Knowledge and Data Engineering.

[4] Nguyen, Trung-Kien, and Yuan Fang. "Diffusion-based Negative Sampling on Graphs for Link Prediction." Proceedings of the ACM on Web Conference 2024.

**Questions:**

Q1. Can the authors clarify the hyperparameter choices for GNNs? Specifically, were parameters optimized for each benchmark setting, and if so, did tuning impact performance rankings? If no such tuning was done, I'm skeptical about the validity of the results.

Q2. What about testing in the inductive setting, where the growth of the network might deviate from its expected behavior?

Q3. Given that the degree-corrected benchmark produces substantial performance shifts, can the authors clarify how this effect arises purely from changing the negative edge sampling process?

Q4. How does the degree-corrected benchmark compare to alternative methods for addressing biases, such as distance-aware benchmarks or methods using degree-aware sampling? Although this is mentioned in the Discussions section, no result has been presented to back this up.

Q5. Could the authors provide additional evidence on the potential overfitting to large-degree nodes within the degree-corrected benchmark? Quantifying this effect with metrics like clustering or centrality could clarify the approach’s limitations.

Q6. Why is Section 2.5 (assuming no degree assortativity) relevant here, given that real-world graphs often exhibit assortative mixing? How does this assumption affect the applicability of your results to real-world graphs?

Q7. Upon simple searches online, I could find other works that alleviate the degree-induced or centrality-induced in GNNs. How do you think this benchmark will complement it? [1] [2]

Q8. I think this sort of study in the Knowledge Graph datasets would be a significant contribution, too, since link prediction is much more prevalent and often complex tasks (such as complex query answering) can be broken down into multiple link prediction tasks naively. The effects, I'm assuming, might be much more profound there. There are similar studies in that domain [3] [4].

[1] Jian Kang, Yan Zhu, Yinglong Xia, Jiebo Luo, and Hanghang Tong. 2022. RawlsGCN: Towards Rawlsian Difference Principle on Graph Convolutional Network. In Proceedings of the ACM Web Conference 2022 (WWW '22).

[2] Arun, Arvindh, et al. "CAFIN: Centrality Aware Fairness Inducing IN-Processing for Unsupervised Representation Learning on Graphs." ECAI 2023.

[3] Kamigaito, Hidetaka, and Katsuhiko Hayashi. "Comprehensive analysis of negative sampling in knowledge graph representation learning." International Conference on Machine Learning. PMLR, 2022.

[4] Madushanka, Tiroshan, and Ryutaro Ichise. "Negative Sampling in Knowledge Graph Representation Learning: A Review." arXiv preprint arXiv:2402.19195 (2024).

---

> ### Author Response · Authors · 2024-11-15
> **Request for clarification**
>
> Thank you for taking the time to review our paper and for providing valuable feedback. We would appreciate clarification on a few specific comments to ensure we address them effectively. We are actively working on the other comments not included here and will submit our responses in the coming week before the deadline.
>
> > W4. The choice of only four GNNs among 26 total methods seems outdated, as classical link prediction methods are now less commonly used. Incorporating a more diverse set of GNN models would better reflect the benchmark’s utility in current graph machine learning applications. Graph Transformers and their variants, which are often the most commonly used ones in practice now, are completely missing from the evaluations.
>
> In response to another referee's suggestion, we are considering integrating state-of-the-art GNNs, specifically BUDDY or NCN. However, we seek clarification on the intent behind this recommendation. Our primary concern is to identify the limitation of link prediction benchmarks, rather than identifying the best link prediction methods. We anticipate that these advanced methods may perform similarly or better than those we've tested. Regardless of the result, our main finding remains the same: positive edges typically form between high-degree nodes and can be differentiated from negative edges based solely on degree. The bias occurs during the generation of the benchmark dataset, prior to applying any link prediction methods. Since no link prediction method affects this process, we are uncertain why incorporating state-of-the-art methods would impact our results and would like to hear the intent of the suggestion to fully address the referee's concern.
>
> > Q2. What about testing in the inductive setting, where the growth of the network might deviate from its expected behavior?
>
> Do you mean testing how the link prediction algorithm performance changes when the generative model of the network changes? Meaning if a network is evolving following a generative model A up until time $t$ (and follows model B thereafter), will a link prediction algorithm trained until time $t$ be able to predict links the network after time $t$? If we understand the question correctly, while this is an interesting question about "temporal" link prediction, we would like to clarify that we focus on mitigating a bias in "missing" link prediction benchmarks, instead of "temporal" link prediction. Are we misinterpreting your question?
>
> > Q5. Could the authors provide additional evidence on the potential overfitting to large-degree nodes within the degree-corrected benchmark? Quantifying this effect with metrics like clustering or centrality could clarify the approach’s limitations.
>
> Could you please clarify what you mean here? Specifically, what do you mean "potential overfitting to large-degree nodes" and how can metrics like "clustering or centrality" quantify this?
> Do you mean we should show the variance of clustering/centrality of nodes in a graph in Fig 1 C/D/E instead of degree heterogeneity? If yes, while we can do this, we don't fully understand the intention behind it.

---

> ### Comment · Reviewer_1S2G · 2024-11-18
> **Clarifying my concerns**
>
> > Our primary concern is to identify the limitation of link prediction benchmarks, rather than identifying the best link prediction methods. We anticipate that these advanced methods may perform similarly or better than those we've tested.
>
> One of the primary claims seems to be that *"... overly reliant on node degree, to the extent that a 'null' method based solely on node degree can nearly match optimal performance"*. What is optimal performance here? If you have not benchmarked the recent state-of-the-art GNNs (like variants of Graph Transformers and rewiring-infused GNNs) with **proper hyperparameter tuning**, I'm unsure how it can be called "optimal". To be precise, the position of PA in Figures 1D and 2A can possibly move down a lot with proper HP tuning for other methods and by adding newer GNN architectures.
>
>
> > while this is an interesting question about "temporal" link prediction, we would like to clarify that we focus on mitigating a bias in "missing" link prediction benchmarks, instead of "temporal" link prediction
>
> Thank you for the clarification; this then warrants an explicit mention that your work focuses only on the transductive setting. Inductive is a much more practical setting since transductive assumes the existence of the whole graph before prediction, which is rarely observed in the real world and hence is moving out of fashion as the standard benchmark currently [1]. The common inductive link prediction setup involves inference (predicting links) on a new, unseen graph (on unseen nodes with unknown degrees), which can possibly change the underlying governing network model, which was what my concern was about - does this degree-corrected benchmark offer any non-trivial advantages or disadvantages in the inductive setting?
>
>
> > Could you please clarify what you mean here? Specifically, what do you mean "potential overfitting to large-degree nodes" and how can metrics like "clustering or centrality" quantify this?
>
> Sure, I suggest quantifying the claim that your degree-corrected benchmark is less susceptible to degree overfitting, which would be great. Using "clustering or centrality" was just a possible suggestion and not a requirement or recommendation from my side. *"... As a result, the model is penalized if it overfits these large-degree nodes. Instead, the model is encouraged to learn salient features of other nodes, resulting in embeddings that better capture nuanced network structures, such as community structure"*. While community detection is correlated to link prediction, I'm unsure if they can be used to make causal claims. Here's the rephrased question -
> 1. While it may make intuitive sense, can you design an experiment to empirically show that the new benchmark does not introduce a new form of degree bias since high-degree nodes are sampled more?
> 2. While it is true that high-degree nodes are also sampled frequently as negative samples, there still exists a case where a low-degree previously had a probability of $\frac{1}{n}$ to be chosen as a negative sample before and now it could be $<< \frac{1}{n}$ depending on the network model. What does it mean for the models to never (with very low probability) see certain (low-degree) nodes in either the positive or negative training sets while previously there were at least present in the negative set? Does this introduce a bias or an issue from the fairness perspective?
>
>
> [1] Renming Liu, Semih Canturk, Frederik Wenkel, Sarah McGuire, Xinyi Wang, Anna Little, Leslie O'Bray, Michael Perlmutter, Bastian Rieck, Matthew Hirn, Guy Wolf, & Ladislav Rampašek (2022). Taxonomy of Benchmarks in Graph Representation Learning. In The First Learning on Graphs Conference.

---

> ### Author Response · Authors · 2024-11-26
>
> > Q1. Can the authors clarify the hyperparameter choices for GNNs? Specifically, were parameters optimized for each benchmark setting, and if so, did tuning impact performance rankings? If no such tuning was done, I'm skeptical about the validity of the results.
>
> First of all, thank you so much for the time for reviews and the constructive suggestion. Before discussing hyperparameters, we would like to emphasize that the fundamental flaw in the link prediction benchmark exists independently of method performance. The core issue is that node degree creates a "shortcut" that even simple methods like PA can exploit to differentiate between positive and negative edges. When methods exploit this shortcut, the benchmark overestimates their performance while potentially underestimating methods that rely more on other meaningful structural features .
> Our results in Section 2.3 demonstrate this concretely: the standard benchmark's rankings poorly align with actual performance in recommendation tasks. While hyperparameter tuning might improve individual method performance, it cannot address this fundamental benchmark flaw that rewards exploitation of a trivial feature rather than learning of meaningful network structure.
> Even if GNNs improve performance by hyperparameter tuning, it does not invalidate none of our claims about the benchmark.
>
> Having said this, we still would like to demonstrate that hyperparameter tuning does not alter our message much. We have used the standard hyperparameter set as they represent the "standard" setup. We now implemented hyperparameter tuning for all GNN methods including a newly added BUDDY link prediction method. We adopt held-out validation to select (i) the number of layers $\{1,2\}$, (ii) the hidden channel size $(\{64,128,256\})$ for each layer. We observed that the selected hyperparameters differ across datasets, as anticipated by the referee. We confirmed that our results are consistent with the previous results (see Figs. 1, 2, and 3 in the revised manuscript).
>
> > Q2. What about testing in the inductive setting, where the growth of the network might deviate from its expected behavior?
>
> In transductive settings, link prediction occurs between nodes in the training graph, while inductive settings involve predicting links for new nodes or unseen graphs. Inductive link prediction often uses similar approaches to transductive ones [A1,A2,A3], such as classifying edges as positive or negative or retrieving the top-k most likely edges.
> Importantly, the degree bias persists in the inductive setting. For example, when sampling edges from new nodes uniformly at random, a node with $k$ new edges is $k$ times more likely to be selected than a node with $1$ new edge, mirroring the degree bias in the transductive setting.
>
> Transductive link prediction is as equally important as inductive ones and should not be neglected; for example, in citation networks, predicting missing citations between existing papers (transductive) is just as crucial as predicting citations for newly published papers (inductive), as it helps discover missing knowledge connections, assess research impact accurately, and maintain citation quality control.
>
> We've added a discussion of this point to the revised manuscript but refrain from making claims about our benchmark's effectiveness in the inductive setting, as this presents an interesting direction for future research.
>
> - [A1] Chen, Jiajun et al. “Topology-Aware Correlations Between Relations for Inductive Link Prediction in Knowledge Graphs.” ArXiv abs/2103.03642 (2021)
> - [A2] Wang, Yanbang et al. “Inductive Representation Learning in Temporal Networks via Causal Anonymous Walks.” ArXiv abs/2101.05974 (2021): n. pag.
> - [A3] Xu, Da, et al. "Inductive representation learning on temporal graphs." arXiv preprint arXiv:2002.07962 (2020).

---

> ### Author Response · Authors · 2024-11-26
>
> > Q3. Given that the degree-corrected benchmark produces substantial performance shifts, can the authors clarify how this effect arises purely from changing the negative edge sampling process?
>
> We isolated the effect of our negative edge sampling process by keeping all other components identical to the standard setup.
> In both link prediction and community detection tasks, negative edge sampling is the sole modified component. Thus, the performance shifts are attributed to this change.
> The modification of negative edge sampling can create secondary effects such as discreasing distances between nodes as we discussed in the discussion section, which, however, stem from the negative edge sampling process.
>
> To further test the effect of our negative edge sampling process, we also train GNNs on the negative samples generated by a distance-aware benchmark (referred as HeaRT). Since HeaRT is computationally expensive, we selectively run the experiment on graphs with well-separated communities.
> While GNNs trained on other benchmarks are successful for these graphs, we find that HeaRT-trained GNNs performed substantially worse as follows:
>
> | Mixing rate | GAT | GCN | GIN | GraphSAGE |
> |-------------|-----|-----|-----|-----------|
> | 0.10 | 0.00668 | 0.04247 | 0.00603 | -0.00063 |
> | 0.15 | 0.01600 | 0.01383 | 0.02049 | -0.00004 |
> | 0.20 | 0.00951 | 0.01670 | 0.00955 | -0.00023 |
>
> The results show that direct distance control by HeaRT did not contribute positively to representation learning. And that the positive effect of our degree-corrected benchmark is not directly due to distance control but due to the correction of the degree bias arising from edge sampling.
>
> > Q4. How does the degree-corrected benchmark compare to alternative methods for addressing biases, such as distance-aware benchmarks or methods using degree-aware sampling? Although this is mentioned in the Discussions section, no result has been presented to back this up.
>
> We now added HeaRT as a new baseline benchmark.
> Our results show that:
> 1. HeaRT provided ranking of methods that is largely different from both the original benchmark and our degree-corrected benchmark.
> 2. The method ranking by HeaRT is the least aligned with that on recommendation tasks. The proposed degree-corrected benchmark remains the best agreement.
>
> > Q5. Could the authors provide additional evidence on the potential overfitting to large-degree nodes within the degree-corrected benchmark? Quantifying this effect with metrics like clustering or centrality could clarify the approach’s limitations.
>
> Thank you for this important question.
> We investigated potential overfitting through a systematic decomposition analysis of the AUC-ROC scores.
> We provided both theoretical and numerical evidences in new Section 3 in the SI in the revised manuscript, and here we provide the overview.
>
> We decompose the AUC-ROC scores into contributions from different node degree groups.
> Namely, we identified that over 70\% of the overall AUC-ROC score contributed by high-degree positive edges and low-degree negative edges in heterogeneous graphs.
> Using a logistic regression model, we confirmed that degree was used as the most important feature, with importance 1.7~2.8 times larger to the second most important feature for degree heterogeneous graphs.
>
> These two results demonstrate that (1) the standard benchmark produces disproportionately high-degree positive edges and low-degree negative edges that are easy to classify with degree, and (2) learning-based methods trained on this benchmark optimize the performance for these overrepresented easy cases, which leads to overfitting.

---

> ### Author Response · Authors · 2024-11-26
>
> > Q6. Why is Section 2.5 (assuming no degree assortativity) relevant here, given that real-world graphs often exhibit assortative mixing? How does this assumption affect the applicability of your results to real-world graphs?
>
> It's important to distinguish between "assortative mixing" and degree assortativity. Assortative mixing is a broad concept where similar nodes tend to connect, while degree assortativity specifically refers to connections between nodes of similar degrees.
> Unlike externally given node features (e.g., gender in social networks), degree assortativity is determined by the network structure itself. Importantly, not all networks exhibit degree assortativity; for example, the Karate club network is a well-known degree-disassortative social network.
>
> We acknowledge that our assumption of no assortativity is simplified. However, our analysis in Section 4.2 in SI demonstrates that degree assortativity has minimal impact on AUC-ROC scores (as shown in Fig. 3A of the SI).
>
> [1] Hu, Hai-Bo, and Xiao-Fan Wang. "Disassortative mixing in online social networks." Europhysics Letters 86.1 (2009): 18003.
>
> > Q7. Upon simple searches online, I could find other works that alleviate the degree-induced or centrality-induced in GNNs. How do you think this benchmark will complement it? [1] [2]
>
> The cited works [1,2] focus on modifying GNN architectures to mitigate degree-based biases, while our work addresses a fundamental flaw in the evaluation framework itself.
> This distinction is crucial.
> Having accurate evaluation metrics is a prerequisite for properly assessing any debiasing method.
>
> Our degree-corrected benchmark could help validate whether architectural solutions are truly effective in real-world tasks. Take RawlsGCN [1] as an example: while it aims for degree-invariant node representations to address inherent degree disparity, our work reveals an artificial bias from edge sampling in the evaluation process. If RawlsGCN shows improved fairness on the standard benchmark, it's unclear whether this improvement comes from addressing real degree disparity or simply compensating for the benchmark's artificial bias. Our degree-corrected benchmark can help disambiguate these effects.
>
>
> > Q8. I think this sort of study in the Knowledge Graph datasets would be a significant contribution, too, since link prediction is much more prevalent and often complex tasks (such as complex query answering) can be broken down into multiple link prediction tasks naively. The effects, I'm assuming, might be much more profound there. There are similar studies in that domain [3] [4].
>
> We would like to clarify that our work addresses bias in evaluation benchmarks, not training.
> While the cited works [3,4] focus on negative sampling during model training, knowledge graph link prediction benchmarks typically evaluate models using uniform random sampling of positive and negative edges, which is the same framework we analyze.
> This means KG benchmarks inherit the degree bias we identify, independent of how models are trained.
> Sophisticated training sampling methods like DMNS [4] are evaluated on benchmarks, and the evaluation can still be skewed by degree bias.
> Thus, the core issue of degree bias in uniform random sampling during evaluation remains fundamental across both simple graphs and KGs.

---

> ### Comment · Reviewer_1S2G · 2024-11-26
> **Final Clarifications**
>
> Thank you for the detailed response. I will go through them and re-evaluate my assessment; however, I still see two of my questions being unaddressed or misunderstood.
>
> > 1. While it may make intuitive sense, can you design an experiment to empirically show that the new benchmark does not introduce a new form of degree bias since high-degree nodes are sampled more?
> 2. While it is true that high-degree nodes are also sampled frequently as negative samples, there still exists a case where a low-degree previously had a probability of $\frac{1}{n}$ to be chosen as a negative sample before and now it could be $<< \frac{1}{n}$ depending on the network model. What does it mean for the models to never (with very low probability) see certain (low-degree) nodes in either the positive or negative training sets while previously there were at least present in the negative set? Does this introduce a bias or an issue from the fairness perspective, or does it not learn any patterns from low-degree nodes?
>
> These two seem to have been skipped over. If it has been explicitly addressed by any of your comments already, please point me to them.
>
> > Thus, the core issue of degree bias in uniform random sampling during evaluation remains fundamental across both simple graphs and KGs.
>
> My intention in bringing KGs into the discussion was not to point out that prior works in that domain have already addressed this bias but rather for the authors to comment on how they hypothesize their newly proposed benchmark would translate to that domain and maybe test a couple of methods as I believe it will supplement that narrative well. Transductive KG completion is a popular task and, on a high level, shares a lot of commonalities with transductive link prediction, and showing this effect on that domain will further necessitate the corrected benchmark.

---

> ### Author Response · Authors · 2024-11-27
>
> We appreciate your prompt response and the opportunity to clarify our revision!
>
> We address the fairness concern in Section 3 of the Supplementary Information (SI) and would like to clarify it here.
>
> In the link prediction task, learning occurs through contrastive learning. The model's objective is to distinguish between positive and negative edges, not between  positive edges and not between negative edges.
> For fairness, we want the model to equally learn four types of comparisons:
> 1. Type 1: high-degree positive vs. high-degree negative,
> 2. Type 2: high-degree positive vs. low-degree negative,
> 3. Type 3: low-degree positive vs. high-degree negative, and
> 4. Type 4: low-degree positive vs. low-degree negative.
>
> Whether the mdoel learns them equally or not depends on the distribution of these four cases, and we identify the distribution in SI Section 3.2.
> Namely, we find that the standard benchmark creates imbalanced distribution and produces disproportionally many type 2 comparisons. In fact, for the heterogenous networks, the contribution of type 2 comparison is over 70% of the AUC-ROC score. On the other hand, our degree-corrected benchmark achieves nearly perfect parity for most networks, i.e., sampling the four types of comparisons nearly equally.
> Thus, our experiment prompted by the referee suggests that it is the standard benchmark that is unfair to the model learning in terms of node degree.
>
> Additionally, we have numerical evidence that the reduced sampling of low-degree nodes in the degree-corrected benchmark does not impair the model learning.
>
> - In the recommendation task (Section 3.2 in the main text), a model is asked to identify the top C=50 nodes in terms of the likelihood of links (estimated by a prediction method) for each node. *The performance is evaluated by the average performance over all nodes*. Since most networks have predominantly many small-degree nodes, the recommendation performance largely reflects the recommendation performance for small-degree nodes. If a benchmark poorly reflects the prediction performance for small-degree nodes, it should poorly correlate with the recommendation task. We find that this is not the case for the degree-corrected benchmark; it aligns the recommendation task better than the standard link prediction benchmark.
>
> - In community detection experiments (Section 3.3 in the main text), degree-corrected training improved detection accuracy across all nodes. When evaluating the performance of community detection, every node is treated equally regardless of its degree and contributes equally to the final detection accuracy. And many nodes have a small degree in the LFR graphs.
> Thus, the community detection performance is largely determined by the classification accuracy for small-degree nodes. If the degree correction impairs the GNNs' learning of small-degree nodes, they would incorrectly place them to be close to wrong communities, resulting in poor community detection performance. We did not find such results in our analysis. Instead, we observed an improvement in the community detection performance by degree-corrected training.
>
> In summary, degree-corrected sampling produces a balanced set of comparisons and is thus fairer than the standard benchmark in terms of node degree. Our experiments across two independent tasks show no evidence that the degree-corrected benchmark impairs learning for small-degree nodes. Rather, we found that it improves both the alignment between link prediction and recommendation tasks, as well as community detection performance.

---

> > ### Comment · Reviewer_1S2G · 2024-12-02
> > **Final remarks**
> >
> > Thank you for the clarifications. They were detailed and addressed most of my concerns. I have increased the final rating and the soundness rating.

---

### Official Review · Reviewer_CfDu · 2024-11-03

**Soundness:** 3
**Presentation:** 4
**Contribution:** 3
**Rating:** 6
**Confidence:** 4

**Summary:**

Observing existing link prediction models often sample partial negative edges for evaluation; this paper hypothesized that this sampled evaluation includes degree bias and would cause the link predictor model to over-fit to capture node degree signal in making a prediction. After empirical and theoretical analysis, this paper successfully demonstrates the degree of bias, proposes a degree unbiased negative sampling method, and demonstrates that the newly proposed benchmark would result in different rankings for some link prediction models and better align with recommender systems.

**Strengths:**

(1) This paper investigates a widely used technique for evaluating link prediction performance. The sampling bias discovered in this technique has never been systematically investigated before.

(2) This paper provides a rigorous justification, not only in terms of theoretical analysis (e.g., derived the relationship between the degree distribution and the PA link prediction AUCROC) but also empirically analyzed the performance.

(3) this paper also demonstrates several implications of using degree-corrected benchmarks, one for aligning with recommendation tasks (which is more aligned with real-world applications) and one for learning community structure.

**Weaknesses:**

(1) When empirically demonstrating the relationship between the degree bias and link prediction performance, this paper only leverages the preferential attachment model, which might limit the generalizability of some observations drawn in the Figure. See Question 1 for more details.

(2) Throughout the analysis, the paper does not leverage any advanced machine learning model for link prediction, such as the more recently proposed BUDDY [1] and NCN [2]. Since both explicitly model the structures into the link prediction decision-making, it will be interesting to see how their performance relates to the degree distribution. Furthermore, the largest graph used in this paper only has up to 100,000 nodes. It might be better to include a further larger graph such as citation2. Moreover, it would be more important to investigate the degree of bias for very large networks since, on small networks, it is very easy to conduct full negative edge link prediction. See question (5) for more suggestions.

[1] Chamberlain, Benjamin Paul, et al. "Graph neural networks for link prediction with subgraph sketching." arXiv preprint arXiv:2209.15486 (2022).
[2] Wang, Xiyuan, Haotong Yang, and Muhan Zhang. "Neural common neighbor with completion for link prediction." arXiv preprint arXiv:2302.00890 (2023).
[3] Hu, Weihua, et al. "Open graph benchmark: Datasets for machine learning on graphs." Advances in neural information processing systems 33 (2020): 22118-22133.

(3) The motivation of the section 3.3 is unclear. If it aims to show that the proposed benchmark captures a lower node degree, would it be better to directly demonstrate that the learned node embedding can lead to better degree prediction performance? If it is to show the benchmark capture more salient graph structures, I wonder if there is any application where the link prediction performance requires capturing the substructures.

**Questions:**

(1) In addition to the PA model that only belongs to the heuristic-based method, I am wondering whether we can design a more explicit method using degree, for example, designing an MLP based on only the node degree and train the model for link prediction compared with designing an MLP not based on degree and train the model for link prediction, and compare their performance. Experimenting with this way could further demonstrate such a degree of bias, which could also be captured by the learning-based method.

(2) Since we will split the original edges into training/validation/testing and we will remove testing edges in both message-passing and setup of supervision edges for training, so we also change our graph and I am wondering whether such graph change would cause the heterogeneity also change? Since following the same logic, for high-degree nodes, their neighboring edges are more easily selected as testing edges and removed

(3) In Figure D, the author attributes the advantages of the PA model over others to its explicit ability to capture degree signals. However, I’m curious about models that outperform PA. What are these models, and do they also capture degree signals? It might be better to analyze those better than PA and derive insights on whether they can capture degree signals.

(4) There are some other references discussing the degree-related bias:
 [1] Wang, Yu, and Tyler Derr. "Degree-related bias in link prediction." 2022 IEEE International Conference on Data Mining Workshops (ICDMW). IEEE, 2022.
 [2] Subramonian, Arjun, Jian Kang, and Yizhou Sun. "Theoretical and Empirical Insights into the Origins of Degree Bias in Graph Neural Networks." arXiv preprint arXiv:2404.03139 (2024).

(5) I am wondering, specifically for large-scale networks, such as the ones with nodes beyond 100,000 nodes, what would the conclusion be? I noticed from Table 2,3,4, currently the most large network is Collab (232865), which is still far less than the number of nodes in the real-world large social network. I am wondering, for those networks, whether the observed degree bias still exists. Since, for small-scale networks, we can very easily conduct full negative edge analysis rather than sample negative edges, it is essential to analyze the large-scale social network.

---

> ### Author Response · Authors · 2024-11-15
> **Request for clarification**
>
> Thank you for taking the time to review our paper and for providing valuable feedback.
> We would appreciate clarification on a few specific comments to ensure we address them effectively.
> We are actively working on the other comments not included here and will submit our responses in the coming week before the deadline.
>
> > (2) Figure D demonstrates that different graphs may have different heterogeneity in their sampled subgraph degree distribution. Is there any relationship between the original graph property and the heterogeneity of the sampled graph?
>
> In Figure 1 D, we do not sample a subgraph to calculate the heterogeneity. We calculate it on the original graph itself. Have we misunderstood your question?

---

> > ### Comment · Reviewer_CfDu · 2024-11-25
> > **Thank you for your clarification**
> >
> > Thank you for clarifying my comment. I have updated it to include more context:
> >
> > Since we will split the original edges into training/validation/testing and we will remove testing edges in both message-passing and setup of supervision edges for training, so we also change our graph and I am wondering whether such graph change would cause the heterogeneity also change? Since following the same logic, for high-degree nodes, their neighboring edges are more easily selected as testing edges and removed

---

> ### Author Response · Authors · 2024-11-26
>
> We thank the reviewer for their thoughtful feedback and suggestions. We appreciate the detailed questions which will help improve the clarity and completeness of our work.
>
> > Questions:
> > (1) In addition to the PA model that only belongs to the heuristic-based method, I am wondering whether we can design a more explicit method using degree, for example, designing an MLP based on only the node degree and train the model for link prediction compared with designing an MLP not based on degree and train the model for link prediction, and compare their performance. Experimenting with this way could further demonstrate such a degree of bias, which could also be captured by the learning-based method.
>
> Thank you for this insightful suggestion. We conducted additional experiments to explicitly quantify how degree information influences learning-based methods.
>
> We designed two MLP models to compare degree-based versus structural features. The first MLP used only degree-related features i.e., individual node degrees, their product, sum, and max/min values. The second MLP used only structural features i.e., resource allocation, Jaccard, Adamic-Adar, and random walk indices. Both models had two hidden layers with LeakyReLU activation and were optimized using held-out validation, testing hidden layer sizes of 32 and 64 with dropout rates of 0.2 and 0.5.
>
> We observed that:
> - The degree-based MLP perform very similarly with the PA (Pearson's correlation > 0.99; the standard deviation of AUC-ROC < 0.01).
> - When the degree distribution is *homogeneous* ($\sigma < 0.8$), the MLP based on degree underperformed the MLP without degree features (AUC-ROC <0.7 for MLP w/ degree and AUC-ROC >0.7 for MLP w/o degree).
> - However, when the degree heterogeneity increases, the MLP with degree improved its performance and ourperformed the MLP without degree almost all cases above $\sigma > 1.5$, which includes the OBG graphs.
>
> This suggests that learning-based methods can be affected by degree bias when they exploit degree information.
>
> We then asked: when both degree-related and other structural features are presented, do learning-based methods still exploit degree information?
> We conducted a controlled experiment using ridge regression to directly quantify the relative importance of degree versus non-degree features through their regression coefficients. Our analysis involved the following steps:
>
> 1. We focused on the degree product $(k_i k_j)$ and all features used in the Non-degree-based MLP
> 2. To eliminate collinearity effects, we performed feature orthogonalization by regressing non-degree features on degree product and used their residuals as new non-degree features
> 3. We scaled the features to have unit L2 norm to make the coefficients directly comparable across different features
> 4. We trained a ridge regression model for the features on the link prediction task
>
> The mean absolute coefficients across different graph types are summarized below:
>
> | Feature | Coefficient (all graphs) | Coefficient (graphs with $\sigma > 1$) | Coefficient (graphs with $\sigma > 1.5$) |
> |---------|------------|------------|------------|
> | Random walk | **11.52** | 10.68 | 9.01 |
> | Degree product | 8.79 | **17.19** | **28.00** |
> | Resource allocation | 2.81 | 4.53 | 6.88 |
> | Adamic-Adar | 1.35 | 2.35 | 2.89 |
> | Jaccard index | 0.89 | 1.06 | 0.17 |
>
> The degree product becomes substantially more important in graphs with heterogeneous degree distributions ($\sigma > 1.0$), with its coefficient more than 1.7~2.8 times larger than the next most important feature.
>
> This demonstrates that learning-based methods can identify degree as an easy "shortcut", achieving high benchmark performance by primarily exploiting degree information rather than learning other structural patterns. This finding further underscores the necessity of our degree-corrected benchmark to prevent such shortcuts and encourage learning of graph structure.
>
> The results are summarized in Section 3 of the SI in the revised manuscript.

---

> ### Author Response · Authors · 2024-11-26
>
> > (2) Figure D demonstrates that different graphs may have different heterogeneity in their sampled subgraph degree distribution. Is there any relationship between the original graph property and the heterogeneity of the sampled graph?
>
> We believe the referee is asking about how removing test edges affects the degree distribution of the graph. This is an excellent question.
> When we remove test edges for evaluation, the degree of each node in the training graph will be slightly lower than in the original graph.
> However, since test edges are sampled uniformly at random, the expected reduction in degree is proportional to the original degree: if $k_i$ is the original degree of node $i$ and $p$ is the fraction of edges used for testing,then the expected degree in the training graph is $(1-p)*k_i$.
> This proportional relationship means that the relative differences between node degrees are preserved, and the degree heterogeneity (measured by $\sigma$ of the log-normal distribution) remains the same (only the location parameter is shifted).
>
> > (3) In Figure D, the author attributes the advantages of the PA model over others to its explicit ability to capture degree signals. However, I’m curious about models that outperform PA. What are these models, and do they also capture degree signals? It might be better to analyze those better than PA and derive insights on whether they can capture degree signals.
>
> Thank you for highlighting the importance of analyzing the models that outperform PA. Upon examination, we found that while these models achieve higher performance, they do not rely on capturing degree signals as explicitly as the PA model. This is evident from the relatively weak correlation between their AUC-ROC scores and degree heterogeneity, unlike PA, whose performance is strongly tied to degree heterogeneity (SI Section 6 in the revised manuscript). These models generally outperform PA even in scenarios with low degree heterogeneity, suggesting their success stems from capturing other structural patterns rather than leveraging degree signals.
>
> > (4) There are some other references discussing the degree-related bias: .<omitted due to character limit>
>
> Thank you for suggesting the relevant papers.
> We now have added citations to Wang & Derr (2022) and Subramonian et al. (2024) and expanded our discussion of related work.
>
> Wang & Derr (2022) focus on evaluation metrics, demonstrating that Recall provides a less biased evaluation compared to metrics like NDCG and Precision for link prediction tasks.
> This work is relevant to our work in that it focuses on bias in the evaluation procedure, though it focuses on metric choice while we focus on the edge sampling procedure.
>
> Subramonian et al. (2024) provide theoretical analysis of how node degree affects GNN performance through message passing and neighborhood structures.
> While they focus on architectural bias in GNNs, our work examines how the edge sampling in benchmark introduces can skew the performance evaluation of the GNNs, instead of training bias.
>
> > (5) I am wondering, specifically for large-scale networks, such as the ones with nodes beyond 100,000 nodes, what would the conclusion be? I noticed from Table 2,3,4, currently the most large network is Collab (232865), which is still far less than the number of nodes in the real-world large social network. I am wondering, for those networks, whether the observed degree bias still exists. Since, for small-scale networks, we can very easily conduct full negative edge analysis rather than sample negative edges, it is essential to analyze the large-scale social network.
>
> This is a valid question. We now tested two citation networks, SciSci and USPTO citation networks.
> The SciSci citation network represents citations between more than 95M publications of all sciences and the USPTO citation network consists of more than 7M patents in the US.
> We believe that these networks are large enough to represent the large-scale real-world networks.
>
> We then followed the same procedure to test the degree bias and measured the AUC-ROC score of the preferential attachment model. We find that the AUC-ROC score is substantially high (0.9452 for SciSci and 0.881 for USPTO), which is in very close agreement with our theoretical prediction (0.9479 and 0.918).
> With the degree-corrected benchmark, the AUC-ROC scores for PA decreases for both graphs, e.g., 0.5018 for SciSci and 0.4818 for the USPTO.
> Thus, the results show the degree bias in the large-scale networks and that the degree-corrected benchmark is effective in mitigating this bias.
> While we do not run other link prediction methods on these networks due to computational constraints, these results provide strong evidence that our findings about degree bias are not limited to smaller networks but represent a fundamental characteristic of the standard link prediction benchmark.

---

### Official Review · Reviewer_hFyf · 2024-11-04

**Soundness:** 2
**Presentation:** 3
**Contribution:** 3
**Rating:** 6
**Confidence:** 4

**Summary:**

This paper is focused on the link prediction task and it shows how the sampling procedure applied in the evaluation of link prediction methods is biased towards high degree nodes. More specifically, the selection of random negative pairs to be distinguished against positive pairs leads to negative pairs connecting lower-degree nodes compared with the positive ones. This issue is analyzed both empirically and theoretically. To address the issue, the paper proposes a degree-correlated sampling procedure that generates negative pairs that have similar degrees as positive ones. Using this new benchmark provides a different ranking of link prediction approaches, most notably with preferential attachment achieving significantly worse results compared with GNN-based methods.

**Strengths:**

- The paper addresses an important issue in the evaluation of link prediction methods.
- The paper is clear and easy to follow.
- The paper applies multiple datasets and methods to support its hypothesis.

**Weaknesses:**

- The definition of unbiased distribution used in the paper needs better justification
- The focus on node degree is a bit narrow given all the possible structural properties that might be biased in negative samples
- The paper does not consider state-of-the-art link prediction methods like BUDDY and NCN.

Detailed comments:

First, I need to disclose that I reviewed this paper earlier, and the authors have improved the paper significantly based on previous reviews. Still, there are some points that I believe could be better addressed:

1) Unbiased distribution: based on the paper, the unbiased degree distribution for disconnected test pairs (no class) should be the same as the one for connected test pairs (yes class). I am still not convinced why that should be the case and how the degree can still be a useful feature in this setting (as mentioned in the paper). For instance, if the proposed sampling algorithm is applied to a Barabasi-Albert graph, how would link prediction algorithms perform? Shouldn't the PA algorithm perform well in this case even if samples are unbiased?

2) Other types of bias: the paper focuses on degree bias but one could expect any connectivity-based metric to also produce a similar bias. The authors also mention shortest paths but there could be many others.

3) Link prediction methods: the proposed link prediction methods are not state-of-the-art, as shown in the papers below:

Chamberlain et al. Graph Neural Networks for Link Prediction with Subgraph Sketching.

Wang et al. Neural Common Neighbor with Completion for Link Prediction.

**Questions:**

1) What is the motivation for the proposed unbiased degree distribution?

2) What other similar sources of bias could affect link prediction performance? Could the proposed approach be generalized?

3) Why state-of-the-art link prediction algorithms were not considered in the experiments?

**Details Of Ethics Concerns:**

I did not identify any ethical concern in the related to this paper.

---

> ### Author Response · Authors · 2024-11-15
> **Request for clarification**
>
> Thank you for taking the time to review our paper and for providing valuable feedback.
> We would appreciate clarification on a few specific comments to ensure we address them effectively.
> We are actively working on the other comments not included here and will submit our responses in the coming week before the deadline.
>
> > Other types of bias: the paper focuses on degree bias but one could expect any connectivity-based metric to also produce a similar bias. The authors also mention shortest paths but there could be many others.
>
> Based on the the reviews we received previously, we have discussed other biases (like the number of common neighbors) in the discussion session and how our proposed benchmarks mitigate these other biases too.
> In this round, we are also comparing our results to a different benchmark which mitigates the differences in distance between the positive and negative sets.
> We have already shown in our discussion that mitigating degree bias reduces the distance bias too but we are now conducting more experiments on the distance debiased benchmark to compare our results.
> It is unclear to us what other types of bias you have in mind. If you can elaborate on the "many others" we can do a better job trying to address them in our final response.
>
> > Link prediction methods: the proposed link prediction methods are not state-of-the-art, as shown in the papers below
>
> We aren't presenting a state-of-the-art link prediction method here. We propose a benchmark to evaluate the link prediction methods. Are we misunderstanding the intent of your comment?

---

> > ### Comment · Reviewer_hFyf · 2024-11-20
> > **Clarification**
> >
> > Here is some clarification as asked by the authors.
> >
> > Other types of bias: my understanding is that the proposed solution addresses other types of bias as a side effect but not as part of its objective, which is focused only on degree. My concern is more regarding why the focus on degree and not other properties, whether the proposed solution could also be applied to reduce other sources of bias explicitly, and whether this has to be done manually or the possible biases can be identified automatically to make sure that the reported comparisons are indeed fair across multiple properties.
> >
> >  Link prediction methods: state-of-the-art methods for link prediction do a much better job at capturing the graph topology than simple GNNs (1-WL-powerful message-passing neural nets). Results on the unbiased setting could change the ranking of these state-of-the-art methods, which would have a significant impact on the ongoing research on link prediction. Otherwise, if these methods still remain at the top, I would say that the impact of this paper is not as significant.

---

> ### Author Response · Authors · 2024-11-26
>
> > Unbiased distribution: based on the paper, the unbiased degree distribution for disconnected test pairs (no class) should be the same as the one for connected test pairs (yes class). I am still not convinced why that should be the case and how the degree can still be a useful feature in this setting (as mentioned in the paper). For instance, if the proposed sampling algorithm is applied to a Barabasi-Albert graph, how would link prediction algorithms perform? Shouldn't the PA algorithm perform well in this case even if samples are unbiased?
>
> We thank the reviewer for continuing to engage with us and for the constructive feedback!
> Let us clarify two key points.
>
> First, regarding why the degree distributions should match, the goal is not to eliminate degree as a predictive feature entirely, but rather to remove the artificial bias introduced by the sampling procedure itself.
> In the standard benchmark, even if the edges form irrespective of degree (e.g., Erdős-Rényi graphs), the sampling method artificially makes high-degree nodes appear more frequently in positive pairs.
> This creates a spurious signal that algorithms can exploit without learning anything about the graph's actual structure.
> By sampling negative pairs with the same degree distribution as positive pairs, we prevent models from exploiting this spurious signal and make the benchmark a more reliable test of whether methods are learning underlying graph structures.
>
> Second, regarding PA's performance on Barabasi-Albert graphs, the reviewer raises an excellent point.
> This echoes a very important constructive feedback we received in the previous reviews, and we have spelled out in Section 3.1 that our goal is "not to completely eliminate degree as a predictive feature, but rather to remove the bias introduced by negative edge sampling." The 'biokg drug' graph example in our results demonstrates this: PA maintains meaningful predictive power (AUC-ROC >> 0.5) when the degree truly correlates with edge probability.
>
> Now, let us focus on the case the referee is concerned about,  where degree is predictive of edges, as represented by the 'biokg drug' graph.
> For such graphs, we ask: does degree-based link prediction methods still perform well under our degree-corrected benchmark?
> In addition to the PA results already presented, we trained a multilayer perceptron (MLP) using only degree-related information as input (e.g., degree sum $k_i + k_j$, degree product $k_i k_j$, minimum degree $\text{min}(k_i, k_j)$ and maximum degree $\text{max}(k_i,k_j)$).
> The MLP can still achieve significantly high AUC-ROC of 0.90 for `ogbl-biokg_drug`, suggesting that degree remains a useful feature under our degree-corrected benchmark.
>
> These results indicate that the degree-corrected benchmark does not completely wipe out degree information. Instead, it targets the degree bias arising from the sampling bias. Simple learning-based models (logistic regression model and MLP) can identify degree as an important feature when it is predictive of edges.

---

> ### Author Response · Authors · 2024-11-26
>
> > Other types of bias: the paper focuses on degree bias but one could expect any connectivity-based metric to also produce a similar bias. The authors also mention shortest paths but there could be many others.
>
> While multiple types of biases can affect link prediction benchmarks, our analysis reveals an interesting hierarchy among them.
> Prompted by the referees, we have implemented the distance-based benchmark proposed by Li et al. (NeurIPS 2024).
> We found that its performance evaluations showed weaker alignment with real recommendation tasks compared to our degree-corrected benchmark (see Figure 3 in the revised manuscript).
>
> More importantly, we observed an asymmetric relationship between degree and distance biases: while our degree-corrected sampling naturally mitigates distance bias (as evidenced by more negative edges connected by paths of length 2), distance debiasing fails to address degree bias effectively, as evidenced by the fact that PA, a purely degree-based predictor, remains the top performer even under distance-bias correction (Fig. 2 in SI).
>
> This asymmetry likely stems from the fact that node distances are inherently influenced by degree heterogeneity.
> Consider a simple example: in a network with high degree heterogeneity, high-degree nodes act as hubs that create short paths between many node pairs.
> When we correct for degree bias, we naturally reduce the effect of this short paths intermediated by high-degree nodes.
> However, correcting for distance alone does not address the underlying degree heterogeneity.
>
> We again emphasize that our focus is on the degree bias introduced by the sampling procedure itself. This point is crucial: while other connectivity metrics like shortest paths can indeed introduce biases, these biases stem from actual structural properties of the networks.
> In contrast, the degree bias we address is purely an artifact of how we sample edges for evaluation.
> This explains why this sampling-induced degree bias is fundamental; it systematically distorts evaluation regardless of the underlying network structure unless the graph is a regular graph.
>
> > Link prediction methods: the proposed link prediction methods are not state-of-the-art, as shown in the papers below:
> >
> > Chamberlain et al. Graph Neural Networks for Link Prediction with Subgraph Sketching.
> > Wang et al. Neural Common Neighbor with Completion for Link Prediction.
>
> We acknowledge that we did not include some state-of-the-art methods and have added BUDDY to the experimental section. While BUDDY showed strong performance on some graphs, its average performance across our test set was not as significantly higher than other methods, even using the original implementation and hyperparameter tuning. This aligns with "no free lunch theorem" that method performance can vary significantly across different graph types and structures, i.e.,  what works well for one class of graphs may not generalize equally well to others.
>
> We would like to re-emphasize that our focus was not on achieving the best possible link prediction performance but rather on understanding a fundamental bias in the evaluation framework itself.
> Having accurate evaluation metrics is a prerequisite for properly assessing any link prediction method, including the state-of-the-art ones.
> Under the standard benchmark, a method can achieve impressive performance without learning meaningful structural patterns.
> Our degree-corrected benchmark could help validate whether such methods are truly effective in real-world tasks or are just exploiting an easy "shortcut" by primarily exploiting degree information rather than learning other structural patterns.

---

> > ### Comment · Reviewer_hFyf · 2024-11-27
> > **Response to rebuttal**
> >
> > Thanks for following up on my comments.
> >
> > I can't find the PA results specifically for ogbl-biokg_drug in the paper. I still believe that the best way to clarify this is to show results for a BA graph where PA should remains one of the best methods but likely not as good as in the standard benchmark.
> >
> > I also think it is confusing to criticize the sampling method because my understanding based on previous answers is that the issue remains even if every pair of nodes is included (i.e. the entire population of pairs is considered).To me, the paper is criticizing what the population should be and that nodes with different degrees should be equally represented in this population. This idea is easier to sell if the goal is to recommend one link per node (then a top-1 correct prediction for a low-degree or a top-degree node is worth the same). This is harder to sell if my goal predict as many missing links as possible in a given graph. In case the authors can convey that the degree-corrected sampling is better even if the goal is to recover something close to the full graph (meaning as many correct edges as possible), that would be very interesting.
> >
> > Overall, even after reading this paper multiple times, the idea in the paragraph above is closest I have been to formally defining what the unbiased distribution of pairs should be.
> >
> > I also found this older paper that seems to relevant and is not cited:
> >
> > Belth et al. A Hidden Challenge of Link Prediction: Which Pairs to Check? ICDM'20.

---

> > > ### Author Response · Authors · 2024-11-30
> > >
> > > We very much appreciate the reviewer's continued engagement and constructive feedback.
> > > We find that the reviewer's concern is valid and makes us think more deeply about the problem.
> > > Thank you for your continued engagement and interaction with us!
> > >
> > > To clarify our results better, we will add the following discussion in the Discussion section and in the SI.
> > >
> > > First, we would like to first clarify the results on ogbl-biokg_drug. Here is the results for PA on this dataset.
> > >
> > > | Negative Edge Sampler | Score |
> > > |----------------------|--------|
> > > | Standard             | 0.998  |
> > > | HeaRT    | 0.771  |
> > > | Degree-corrected | 0.902  |
> > >
> > > The results show that PA still performs well under our degree-corrected benchmark, with AUC-ROC of 0.902.
> > >
> > > Second, regarding the Barabasi-Albert (BA) network results, we have now run experiments on BA networks (3000 nodes and 10 new edges per node) with 10 runs using the degree-corrected benchmark. The following table shows the results for the top 10 methods.
> > >
> > > | Model                  |   Mean AUC-ROC |   Standard deviation |
> > > |:-----------------------|------------------:|-----------------:|
> > > | Buddy                  |             0.591 |            0.007 |
> > > | PA |             0.582 |            0.006 |
> > > | LRW  |             0.578 |            0.005 |
> > > | LPI |             0.533 |            0.004 |
> > > | RA |             0.523 |            0.004 |
> > > | Exp-A |             0.509 |            0.006 |
> > > | vN-A |             0.507 |            0.007 |
> > > | dcSBM                  |             0.498 |            0.004 |
> > > | Exp-L |             0.498 |            0.004 |
> > > | GAT |             0.497 |            0.011 |
> > >
> > > The results show that PA maintains meaningful predictive power (AUC-ROC = 0.582 ± 0.006) under our degree-corrected benchmark, performing second only to BUDDY (0.591 ± 0.007). This aligns with the reviewer's expectation: when degree genuinely correlates with edge formation probability (as in BA networks), degree-based methods should remain effective even under degree-corrected sampling.
> > >
> > >
> > > Third, regarding population vs. sampling, it is crucial to distinguish the following two effects: (1) the genuine mechanism of missing edges and (2) the degree bias in the "missing edges" sampled from the population (e.g., all positive edges).
> > > It is crucial to recognize that the "missing" edges in link prediction benchmarks are artificially generated for evaluation purposes; they don't necessarily represent actual missing edges in the real network.
> > > Our critique focuses on this second type of bias, i.e., the artificial one introduced by test set creation.
> > > This distinction is crucial because otherwise, one may reach incorrect conclusions about the edge formation mechanisms and develop suboptimal methods for real applications.
> > >
> > > Let us illustrate this with a concrete example. Consider developing a citation recommendation system for patent filing based on citation networks. Under the standard benchmark, PA achieves a remarkably high AUC-ROC of 0.88 on USPTO, which might suggest that highly-cited patents tend to cite each other. However, this conclusion would be incorrect; patent citations are primarily driven by technical relevance, field similarity, and temporal ordering, not just the cumulative advantage of popularity (e.g., citation count). The high performance of PA comes from an artificial bias in how we sample test edges, not from capturing real citation mechanisms. This becomes clear when we apply our degree-corrected benchmark: PA's performance drops dramatically to AUC-ROC = 0.48, revealing that it fails to capture meaningful citation patterns.
> > >
> > > The fundamental issue arises not from how we sample negative edges or whether we consider all possible node pairs, but from how we create our test set of "missing" edges.
> > > The "missing" edges in our evaluation aren't truly missing; they're existing edges that we deliberately hide. The way we sample these edges introduces a degree bias that doesn't necessarily reflect real missing links in the network.
> > > This bias persists regardless of how we handle negative edges or whether we consider the full population of node pairs, as long as we sample the "missing" edges uniformly from existing edges for evaluation purposes.

---

> > > > ### Author Response · Authors · 2024-11-30
> > > >
> > > > Thank you for bringing the Belth et al. paper to our attention. Their work addresses the crucial practical challenge of "where to look for edges" in graphs, as searching every pair of nodes is highly inefficient and often infeasible for large graphs. They propose using stochastic block models (SBMs) to limit the search to plausible candidates. Notably, they found that grouping nodes by degree was particularly effective at identifying missing edges.
> > > >
> > > > This result empirically demonstrates that missing edges tend to appear between high-degree nodes. Our work provides a potential mechanism behind this observation, i.e., the degree bias inherent in the standard edge sampling procedure. Since Belth et al. use this same approach (removing a fraction of edges uniformly) to create their benchmark dataset, their findings are influenced by this sampling bias.
> > > >
> > > > This raises a crucial question: are high-degree nodes truly more likely to be connected, or is this observation partially an artifact of degree bias in the edge sampling method? Our work helps researchers distinguish between actual and artifactual effects, allowing them to focus on genuine mechanisms of edge formation when developing link prediction methods.

---

### Author Response · Authors · 2024-11-26
**Response to reviewers**

We thank all reviewers for their thoughtful and constructive feedback!
Their comments have helped us clarify our contributions and strengthen our analysis.

We would like to begin by highlighting the key contribution of our work: we identify that the standard link prediction benchmark has an inherent artificial bias arising from the edge sampling process, making node degree a surprisingly strong link predictor.
Importantly, this bias arises directly from the edge sampling procedure itself, i.e., nodes with higher degrees appear more frequently in positive samples while negative samples lack this degree bias.
This creates a systematic discrepancy that exists **independently of any particular method**.
Even if some methods, when properly tuned, can learn both degree and meaningful structural features, the benchmark itself remains fundamentally biased in its evaluation setup.
This observation has important implications for how we assess and train graph learning methods. With this in mind, we have made several important improvements to our manuscript:

We have substantially expanded our analysis in the following ways:
- Added comprehensive AUC-ROC decomposition analysis showing that in heterogeneous networks, high-degree positive edges and low-degree negative edges account for over 70% of the overall AUC-ROC score (Section 3 in the SI)
- Conducted feature importance analysis using logistic regression, demonstrating that degree becomes the dominant feature (1.7-2.8 times more important than other features) in heterogeneous networks (Section 3 in the SI)
- Included BUDDY in our evaluations and fine-tuned all GNNs using held-out validation (Section 1.2.3 in the SI)
- Added HeaRT (distance-aware benchmark) as a baseline, revealing an interesting asymmetry: while our degree-corrected sampling naturally mitigates distance bias, distance debiasing does not effectively address degree bias, as evidenced by PA being the strongest performer under distance-bias correction (Section 3.2 in the main text)
- Tested our findings on large-scale networks (Section 4.4 in the SI)
- Expanded our discussion of how degree-corrected sampling relates to other biases in link prediction benchmarks (Section 6 in the SI).

We have also made the following corrections:
- The figure 1D and 1E in the previous manuscript presents the results based on Hits@100 by a mistake. We have corrected Figures 1D and 1E to show AUC-ROC scores instead of Hits@100. We apologize for the confusion.


The revised manuscript and supplementary materials incorporate all these changes. We hope that these improvements provide a more comprehensive and rigorous analysis of degree bias in link prediction benchmarks.

---

### Meta-Review · Area_Chair_bv1e · 2024-12-17

**Metareview:**

This paper discusses an important problem of how link prediction evaluation is defined in the context of graph representation learning research.  The authors note that common methods for evaluating link prediction settings are flawed because the asymmetry between degree-biased positive sampling and random negative sampling biases towards high degree nodes, which results in a skewed evaluation which favors methods which rely on node degree.  The authors propose a degree-corrected benchmark in response.

This paper was evaluated as borderline, and unfortunately leaned towards rejection over the course of discussion.  The paper is also fortunate to have several diligent reviewers who gave very detailed feedback and engaged with the authors throughout the period.  There were a few key issues which were raised during the rebuttal and discussion period:

- The authors' focus on degree bias is a good one, but not a new one.  Several prior works have mentioned similar challenges as discussed with multiple reviewers (hFyf, CfDu), and the authors also missed a very relevant graph link prediction benchmark from NeurIPS'23 at the time of submission, as well as another paper from ICDM'20 which hindered the positioning of this work and made the value-add less clear.

- Some reviewers still lack clarity on exactly what the ideal unbiased distribution of pairs of a link prediction benchmark should have is, despite multiple iterations of reading and reviewing this paper (hFyf)

- Several reviewers raised questions around the applicability and relevance of the benchmark to other types of bias (hFyf, CfDu), which may be influenced as the degree is corrected.

I encourage the authors to fold this feedback into the revision.  In particular, the better clarity around positioning of this work wrt existing graph benchmarks focusing on this topic as well as the previous citations would be important for a fresh reader of the paper.

**Additional Comments On Reviewer Discussion:**

Authors engaged with the reviewers concerns during rebuttal and discussion phase.  They added AUC-ROC decomposition analysis, feature importance analysis via logistic regression showing degree is very impactful, new SOTA GNN methods inside the evaluation, new baselines (HeaRT), and discussion of the interplay between degree-corrected sampling in the context of other link prediction benchmarks. Ultimately, reviewers mostly stayed on the borderline, and the most positive reviewer had a lighter review which did not get into the key flaws which multiple other reviewers raised.

---

### Decision · Program_Chairs · 2025-01-22

Reject